# A critical period of neuronal activity results in aberrant neurogenesis rewiring hippocampal circuitry in a mouse model of epilepsy

Zane R. Lybrand[1,2,3], Sonal Goswami[1,2], Jingfei Zhu[4], Veronica Jarzabek[1,2], Nikolas Merlock[1,2], Mahafuza Aktar[4], Courtney Smith[1,2], Ling Zhang[4], Parul Varma[1,2], Kyung-Ok Cho [5,6], Shaoyu Ge[7] & Jenny Hsieh [1,2 ✉]

In the mammalian hippocampus, adult-born granule cells (abGCs) contribute to the function of the dentate gyrus (DG). Disruption of the DG circuitry causes spontaneous recurrent seizures (SRS), which can lead to epilepsy. Although abGCs contribute to local inhibitory feedback circuitry, whether they are involved in epileptogenesis remains elusive. Here, we identify a critical window of activity associated with the aberrant maturation of abGCs characterized by abnormal dendrite morphology, ectopic migration, and SRS. Importantly, in a mouse model of temporal lobe epilepsy, silencing aberrant abGCs during this critical period reduces abnormal dendrite morphology, cell migration, and SRS. Using mono-synaptic tracers, we show silencing aberrant abGCs decreases recurrent CA3 back-projections and restores proper cortical connections to the hippocampus. Furthermore, we show that GABA-mediated amplification of intracellular calcium regulates the early critical period of activity. Our results demonstrate that aberrant neurogenesis rewires hippocampal circuitry aggravating epilepsy in mice.

[1] Department of Biology, The University of Texas at San Antonio, San Antonio, TX, USA. [2] Brain Health Consortium, The University of Texas at San Antonio, San Antonio, TX, USA. [3] Department of Biology, Texas Woman's University, Denton, TX, USA. [4] Department of Molecular Biology, UT Southwestern Medical Center, Dallas, TX, USA. [5] Department of Pharmacology, Catholic Neuroscience Institute, College of Medicine, The Catholic University of Korea, Seoul, South Korea. [6] Department of Biomedicine & Health Sciences, Institute of Aging and Metabolic Diseases, College of Medicine, The Catholic University of Korea, Seoul, South Korea. [7] Department of Neurobiology & Behavior, Stony Brook University, Stony Brook, NY, USA. ✉email: jenny.hsieh@utsa.edu

A primary function of the hippocampus is to process neuronal activity from the cortex essential for episodic memory, spatial learning, pattern recognition, and emotional behavior[1,2]. Circuitry of the dentate gyrus (DG), the principle gating structure of the hippocampus, establishes an inhibitory feedback circuit comprised of interneuron microcircuits that suppress the firing of dentate granule cells (GCs). Unique to this circuit is the continuous production and integration of adult-born GCs (abGCs) from neural stem cells in the subgranular zone (SGZ) of the DG[3,4]. The maturation and proper integration of newborn neurons is regulated by extrinsic cues (growth factors, hormones, disease, and cell contact)[5,6], as well as intrinsic factors (transcription factors, epigenetic modifiers, and neuronal activity)[7–9] to maintain the feedback inhibition that upholds gating function. Brain trauma, infection, or status epilepticus (SE) causes abGCs to mature abnormally taking on aberrant morphology (e.g., ectopic migration, altered dendrites, mossy fiber sprouting, and accelerated maturation) that are thought to generate hyperexcitable cells, and a subsequent breakdown of the dentate gate allowing for unregulated activity to perpetuate recurrent seizures[10–13]. Previously, we used a transgenic approach to ablate aberrant neurogenesis to reduce spontaneous seizures and normalize cognitive deficits associated with epilepsy[14]. What drives aberrant neurogenesis and the pathological influence abGCs have on existing hippocampus circuitry remains an open question. Understanding mechanisms of aberrant neurogenesis and the influence abGCs have on shaping hippocampus circuitry will provide a context for broadly understanding the pathogenesis of neurological and seizure disorders.

To understand pathological mechanisms of aberrant neurogenesis we used chemogenetics to investigate an activity-dependent dysregulation of abGC maturation that disrupts the protective function of the DG. We conclude that activity early in abGC maturation generates aberrant neurons by disrupting migration. These neurons rewire cortical inputs to the hippocampus circuitry that is associated with spontaneous seizures. Finally, we demonstrate aberrant neurogenesis is characterized by an amplified baseline of intracellular calcium in immature abGCs. This amplified calcium is GABA$_A$ receptor dependent and can be modulated with chemogenetics. Regulation of intrinsic calcium is necessary to prevent improper integration of neurons into extant circuitry and is imperative to understanding the pathological genesis of neurological and seizure disorders.

## Results

### Early activation of immature abGCs promotes aberrant neurogenesis.
Hilar ectopic granule cells (EGCs) are a pathological hallmark of disrupted hippocampus function. These cells are known to be hyperexcitable in both rodent models of temporal lobe epilepsy (TLE), and in resected tissue from epilepsy patients[15,16]. Improper migration of abGCs has been linked to hyperactivation of the mTOR pathway[17] and increased intrinsic neuronal activity[18]. Immature abGCs initially receive a depolarizing GABA signal[9], which prior to synapse formation generates a GABA current necessary for their proper development[7]. GABAergic interneurons, parvalbumin (PV), and somatostatin (SST), are the primary source of extrasynaptic GABA within the DG and in mouse epilepsy models[19] and human patients. These interneurons presumably die via excitotoxicity leading to excess GABA in the neurogenic niche. Furthermore, surviving interneurons may compensate for this loss and become hyperexcitable. Due to GABA's depolarizing effect on immature abGCs, this would impose a hyperexcitable signal to maturing abGCs[20].

To determine if activity during stages of immature abGC maturation is sufficient to drive aberrant changes we used in vivo chemogenetics (e.g., DREADDs) to model this persistent hyperexcitable signal by activating abGCs at different stages of maturation. Retrovirus was injected into the DG to express a designer receptor (hM3Dq) in immature abGCs. The synthetic ligand clozapine-N-oxide (CNO; 1 mg/kg) was injected once daily (i.p.) to activate abGCs for the first (0–1w) or second (1–2w) week post-infection to test the effects of chronic activity. Once the neurons were mature (8 wpi), we performed video-EEG for 2 weeks and perfused the mice to study persistent morphological changes (Fig. 1a). Activation during both the 0–1w and 1–2w period of maturation promoted abnormal migration of abGCs into the hilus (Fig. 1b–h). However, there was no change in dendrite development measured in dendrite complexity or angle of primary dendrite (Fig. S1). Thus early neuronal activity is sufficient to promote hilar migration of abGCs.

### Early activation of abGCs is associated with seizure development.
Generalized spontaneous recurrent seizures (SRS), recorded from the cortex, indicate a dysfunctional dentate gate in TLE[21,22]. To determine if abnormal abGC migration can alter circuit-level function, we implanted mice that received hM3Dq activation (0–1w and 1–2w) for video-EEG monitoring 8 weeks after virus infection (Fig. 1i, j). 0–1w activity generated SRS in 60% of hM3Dq injected mice given CNO compared to the vehicle and GFP controls (Fig. 1k). Activation at 1–2w also generated seizures in 80% of hM3Dq-injected mice (Fig. 1k). Thus, aberrant activity during initial stages of maturation is sufficient to promote pathological development of abGCs and is associated with the presence of spontaneous seizures.

### Silencing aberrant abGCs reduces epilepsy pathology and seizures.
We next asked if silencing immature aberrant abGCs in an epilepsy model could prevent the mismigration of abGCs and occurrence of SRS. We used a retrovirus to express the inhibitory DREADD receptor, hM4Di, or a control GFP virus in abGCs (Fig. 2a, b). Following two days of hippocampus virus injections, mice were given pilocarpine (pilo) to induce SE. CNO or vehicle was administered for 2 weeks following pilo to silence aberrant abGC activity. Eight weeks post-pilo, mice were implanted for 2 weeks of video-EEG monitoring then perfused for histology. In the non-epileptic mice (sham), most abGCs clustered around the SGZ and silencing immature abGCs showed no changes in migration (Fig. 2c). In mice that received pilo, the distribution in granule cell migration increased (Fig. 2b–e) and silencing aberrant neurogenesis reduced the number of abGCs in the hilus (Fig. 2d, e). Pilo also shifted the angle of the primary dendrites from around 94° in sham to about 110° in pilo mice, likely preventing appropriate afferent cortical connections (Fig. S2C, D). Silencing aberrant abGCs after SE normalized the angle of the primary dendrite (Fig. 2f–k). However, in sham mice treated with CNO we observed no change in primary dendrite angle (Fig. S2A, B, E–H). In addition, silencing normal neurogenesis in sham mice did shift the branching distally from the soma compared to controls, however this was not observed in pilo mice (Fig. S3). Thus, transiently silencing this cohort of aberrant abGCs corrected their epileptic morphology. Furthermore, pilo mice injected with the hM4Di and treated with CNO showed a reduction in the number of SRS compared to control mice (Fig. 2l, m) but had no change in seizure duration (Fig. 2n). Thus, early intrinsic activity of immature abGCs is sufficient for pathological changes that modulates seizure development in the pilo epilepsy model.

To determine if the observed aberrant morphology is a consequence of abnormal activity, we measured aberrant abGC migration and dendritic angle at 2 weeks post-pilo following CNO administration (Fig. S4A, B). Interestingly, abnormal

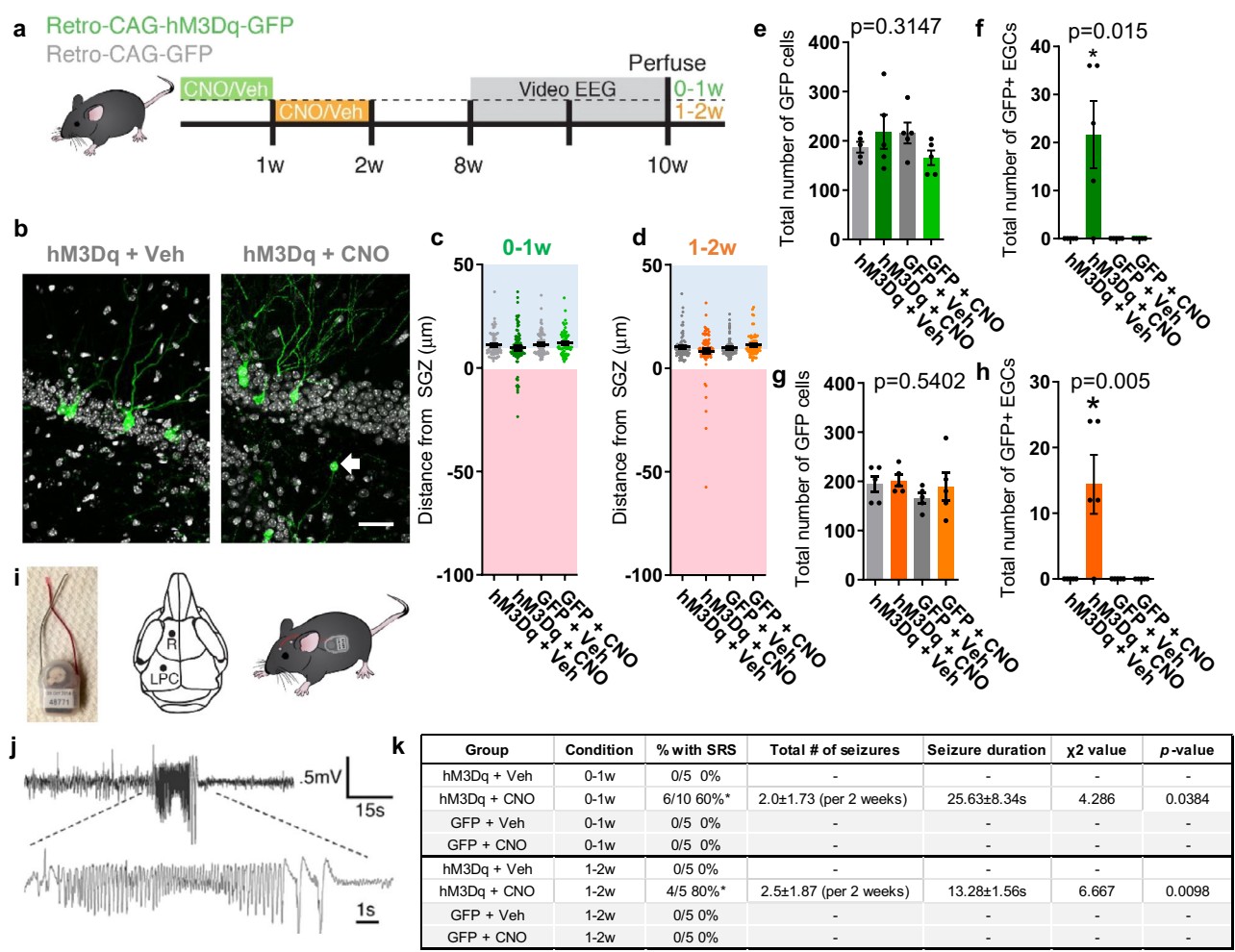

**Fig. 1 Activation of immature adult-born granule cells promote aberrant neurogenesis and seizures. a** Retrovirus was injected into the hippocampus to express the activating chemogenetic receptor (CAG-hM3Dq-GFP) to infect the neural stem cells. Clozapine-N-oxide (1 mg/kg) was administered daily to activate maturing granule cells for different periods of time. At 8 weeks post-infection, mice were implanted for video-EEG to monitor for seizures and then perfused for histology. **b** Representative images of an ectopic granule cell located in the hilus. Arrow indicates cell body. Scale bar represents 50 μm. **c** Quantified migration distance of GFP + cells activated during the first week of maturation (0–1w; green; n = 78, 91, 90, and 69 cells per group, respectively, from five mice per group). **d** Migration distance of GFP + cells activated from the second week (1–2w; orange; n = 81, 84, 79, 69 cells per group, respectively, from five mice per group). Data points represent individual cells. Zero represents SGZ (white), less than zero represents hilus (red), and above ten represents GCL (blue). **e** Quantification of the total number of GFP + cells in the 0–1w group. Gray bars indicate vehicle treatment (Veh). Green bars indicate CNO treatment. n = 5 mice per group. **f** Quantification of the GFP + EGCs in 0–1w group. n = 5 mice per group. **g** Quantification of total number of GFP + cells in 1–2w group. Gray bars indicate Veh and orange indicate CNO treatment. n = 5 mice per group. **h** Quantification of GFP + EGCs in 1–2w group. n = 5 mice per group. *p < 0.05, ANOVA with multiple comparisons. **i** Mice were implanted with wireless EEG transmitters (left) that implant subcutaneously (right). Electrode coordinates (center). **j** Representative trace of cortical seizure. **k** Table of seizure outcomes show that mice activated 0–1w and 1–2w both develop spontaneous seizures compared to GFP and CNO controls. *p < 0.0001, Chi-squared test. Error bars denote SEM; n = 5 and 10 mice per group as noted in the table. All statistics calculated using two-tailed test.

migration and a shift in the primary dendrite angle were already observed at only 2 weeks post-pilo (Fig. S4C, D). When silenced, there was a modest reduction in the number of EGCs in the hilus (Fig. S4A–C), however, at this stage of maturation, the dendrite angle was unaffected (Fig. S4D–F). Therefore, these aberrant morphologies are not observed until after an extended period of time, greater than 2 weeks, and suggests that early abnormal activity is associated with morphology changes. These changes in dendrite morphology seen in mature aberrant abGCs could be an epiphenomenon found in older GCs and may not be causally related to seizures.

**Intrinsic activity regulates aberrant neurogenesis during an early critical period.** Next, we examined whether these changes caused by intrinsic activity are restricted to a developmental window.

To do so, we injected retroviral hM3Dq into non-epileptic mice and gave CNO 8 weeks later once the abGCs were fully matured (Fig. 3a). Acute activation of mature abGCs had no effect on migration, dendrite complexity, nor primary dendritic angle (Fig. 3b–i). Interestingly, when compared to baseline, activation of hM3Dq in mature abGCs showed no difference in SRS measured with EEG (Fig. 3j). Conversely, to determine if silencing mature aberrant abGCs similarly altered circuit-level activity, CNO was administered 6 weeks after SE while recording video-EEG. CNO produced no acute change in SRS activity (Fig. 3k, l) or in ictal spiking activity from baseline in the first 24 h of CNO (Fig. 3m). Chronic CNO showed a modest increase in SRS frequency and seizure duration (Fig. 3p, n, o) suggesting that inhibiting mature abGCs in epilepsy may worsen seizure outcome. Further, no changes in cell morphology were measured by

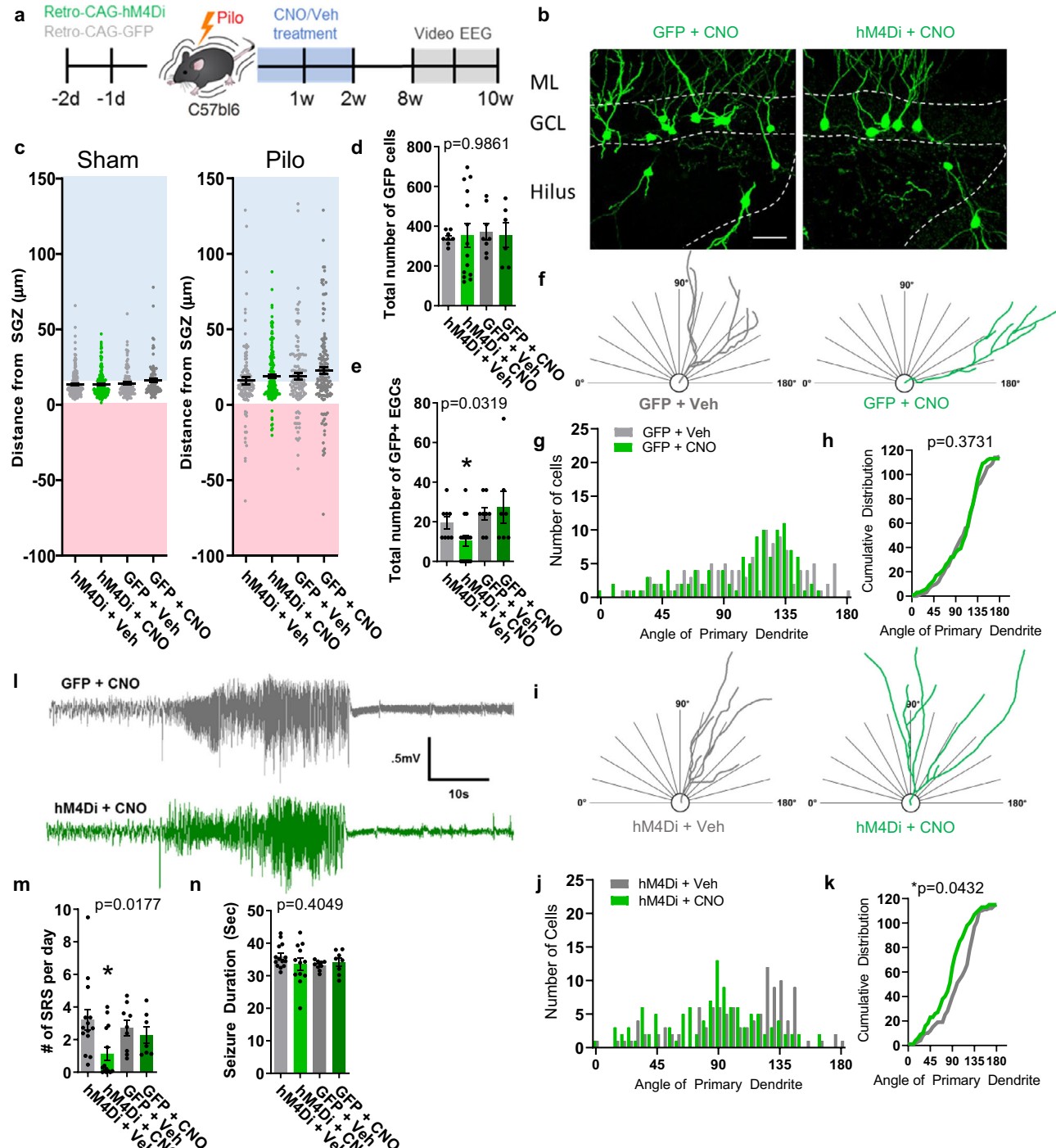

dendrite complexity (Fig. 3q, r) or cell migration and dendrite angle (Fig. S5). Therefore, intrinsic activity regulates aberrant neurogenesis only during an early period of abGC maturation.

**Silencing immature abGCs after pilocarpine treatment prevents the loss of cortical inputs and formation of CA3 back projections**. The protective gating function of the DG is sustained by inhibition from PV and SST interneurons onto abGCs[23]. Breakdown of the gate may be due to a loss of these inhibitory cells accompanied by a gain of recurrent excitatory activity onto dentate abGCs[24]. To determine how aberrant abGCs alter this gating circuitry we used monosynaptic rabies tracing in combination with hM4Di to determine changes to neuronal input of

abGCs from sham mice or aberrant abGCs from SE-induced mice. Mice were injected with either a retro-hM4Di-GFP or control retro-GFP that express a cytoplasmic GFP. They were simultaneously injected with a retroviral avian rabies receptor (TVA) that expresses a nuclear localized GFP (Fig. 4a–c). Two days after the retrovirus, pilo or saline was injected to induce SE or sham and 2 weeks of CNO administration was given to silence immature abGCs. At 8 weeks, mice were injected with Rabies-EnvA-ΔG-mCherry (ΔG-Rb-mCh) to label inputs of abGCs. Only cells expressing TVA were co-infected with ΔG-Rb-mCh (Fig. S6). Cells coexpressing the nuclear TVA and either cytoplasmic hM4Di or GFP were counted (Fig. 4b–d). As expected, pilo increased neurogenesis as we observed a modest increase in

**Fig. 2 Silencing immature adult-born granule cells in pilocarpine model prevents abnormal maturation. a** Schematic of epilepsy model. Retrovirus hM4Di-GFP (green), or the control CAG-GFP (gray), was injected for two consecutive days then given pilocarpine (pilo) to induce the status epilepticus model for epilepsy. Two weeks following pilo, clozapine-N-oxide (CNO) was administered twice daily to silence immature abGCs. Mice were implanted for video-EEG recording and monitored from 8 to 10 weeks post-pilo for SRS and perfused for histology following EEG monitoring. **b** Representative images of the dentate gyrus from epileptic mice. Scale bar equal 50 μm. **c** Migration distance of GFP + cells from the SGZ in sham (left; $n = 207$, 192, 122, and 102 cells per group, respectively, from 15 mice per group) and pilo (right; $n = 187$, 147, 115, and 126 cells per group, respectively, from 8, 8, 6, and 14 mice per group). Each dot represents an individual cell. **d** Quantification of total number of GFP cells from pilo group. **e** Quantification for the total number of GFP + cells located in the hilus. For (**d**) and (**e**), $n = 8$, 8, 6, and 14 mice per group, respectively. Error bars represent SEM, *$p < 0.05$, Student's $t$ test. **f** Representative traces from primary dendrites in GFP group **g** Histogram of the primary dendrite angle for GFP group. Normality (GFP + Veh $p = 0.0802$, K2 = 5.046; GFP + CNO $p = 0.0017$, K2 = 12.76). **h** Cumulative distribution of the primary dendrite angle. *$p < 0.01$, Kolmogorov–Smirnov test. **i** Representative angles from hM4Di group. **j** Histogram of dendrite angle from hM4Di. Normality (hM4Di+Veh $p = 0.0347$, K2 = 6.724; GFP + CNO $p = 0.3601$, K2 = 2.043). **k** Cumulative distribution of the primary dendrite angle. *$p < 0.01$, Kolmogorov–Smirnov test. **l** Representative traces of an electrographic seizure. GFP + CNO is in gray and hM4Di + CNO is in green. **m** Quantification of the spontaneous seizure frequency and (**n**) duration. For (**m**) and (**n**) $n = 14$, 14, 9, and 7 mice per group, respectively. *$p < 0.05$, ANOVA with multiple comparisons. Test for normality presented as D'Agostino & Pearson test. All statistics calculated using two-tailed test.

the number of double labeled GFP + cells in pilo mice compared to sham (Fig. 4d). GFP + and mCherry + cells (i.e., yellow) were identified as starter cells (Fig. 4b, c). Between sham and pilo, this population did not significantly change (Fig. 4i). Cells with mCherry were identified as monosynaptic input cells of the starter population. Because the overall number of starter cells did not change, we measured the total number of mCherry + input cells and determined that, between sham and pilo, there was a significant increase in the number of inputs the aberrant abGCs received, suggesting these expanded input connections might underlie the breakdown of the DG gating mechanism (Fig. 4e). Next, we characterized specific inputs to determine if the shift in the balance of inhibitory to excitatory inputs was altered by silencing with hM4Di. Most input cells were neighboring abGCs (Prox1 +, mCh +) (Figs. 4j and S7B). This population did not change between sham and pilo or in response to silencing of aberrant neurogenesis. Previous work demonstrates few inputs to the abGCs come from neighboring mature GCs, and this connection expands in epilepsy models[24,25]. Neither PV nor SST, the primary inhibitory cells in the dentate circuit, showed any significant changes between sham and pilo, although there was a modest decrease in PV interneurons (Figs. 4f and S7C) and a modest increase in SST interneurons (Figs. 4k and S7D). Adult-born GCs receive three major glutamatergic inputs: from the cortex, other GCs, and mossy cells[26]. Mossy cells were identified by the glutamate receptor GluR2 in the hilus (Fig. S7E) and were found only in the sham condition (Fig. 4g). In pilo, no GluR2 positive input cells were observed in the hilus. This is consistent with the observation that mossy cells death is a key epileptic pathological characteristic thought to contribute to breakdown of the dentate gate[27,28]. Consistent with previous literature, in sham, there were limited back projections found from either the CA3 (Figs. 4l and S7G) or CA1 (Figs. 4m and S7F), but following pilo, these recurrent inputs labeled more cells within both of these hippocampal subfields. Interestingly, following silencing with hM4Di only CA3 back projections were rescued (Fig. 4l). The most drastic change were inputs from the entorhinal cortex (eCTX), a primary excitatory input of the hippocampus circuit (Figs. 4h and S7H). Following pilo, this input population was decreased and rescued by silencing with hM4Di. Thus, chemogenetic silencing of aberrant neurogenesis using hM4Di during an early critical period of abGC maturation prevents miswiring of neuronal input to the dentate circuitry by restoring cortical inputs and blocking recurrent back projections from the CA3.

**Enhanced intracellular calcium in aberrant neurogenesis is regulated by GABA.** The cellular mechanisms for aberrant neurogenesis remain elusive, however our DREADD

manipulation of abGCs suggest that activity dependent mechanisms underlie abnormal maturation, migration, and miswiring all associated with the generation of spontaneous seizures. Compared to mature GCs, immature abGCs (<3 weeks old) are functionally "silent" with limited synaptic input[29,30]. They exhibit smaller voltage-gated sodium and potassium currents, and a reduced ability to generate sodium-dependent action potentials[29–32]. However, these immature abGCs express a high density of T-type calcium channels that generate low threshold calcium-spikes that provide potential activity-dependent maturation cues while the GABAergic inhibition is established and synaptic input (both GABA and glutamatergic) forms to sufficiently drive supra-threshold action potentials[31,33,34]. Therefore, to understand activity-dependent mechanisms of aberrant neurogenesis we used the genetically encoded calcium indicator, GCaMP6f, expressed with a retrovirus to monitor live cell calcium activity of 2-week-old abGCs in hippocampal brain slices from sham mice (Fig. 5a). At baseline, 2-week-old abGCs exhibit dynamic calcium fluctuations (Fig. 5b and Supplementary Movie 1). We observed two forms of calcium events. One is characterized by a fast rise time with a slower decay, typical of intracellular calcium transients from action potentials. The other waveform, similarly is characterized by a fast rise time, however, the decay is much longer often appearing sustained for many seconds reminiscent of a tonic GABA signal. To determine if these calcium transients were action potential dependent, a 60 s baseline was recorded for each cell that was followed by 1 μM TTX delivered to the bath and calcium transients were measured for another 60 s (Fig. 5b). There was no change in the total number of events with TTX (Fig. 5c), however there was a significant decrease in the amplitude (Fig. 5d). Interestingly, with TTX application more sustained calcium transients were present but it is unclear if the shorter events disappeared or if the sustained events mask them. Because GABA plays a critical role in immature abGC maturation, we followed TTX application with 20 μM bicuculline, a GABA$_A$ receptor antagonist to determine the GABA$_A$ receptor dependent component to the calcium transients. Bicuculline nearly abolished all spontaneous calcium transients (Fig. 5c, d). Furthermore, in a separate set of slice preparations, we added exogenous GABA and observed an increase in intracellular calcium (Fig. S8A–C and Supplementary Movie 2). This did not change the event frequency (Fig. S8D), but showed a modest reduction in event amplitude (Fig. S8E). In most cells, we observed an elevated baseline calcium, that when washed out returned to normal (Fig. S8F and Supplementary Movie 3). This elevated calcium baseline likely accounts for the reduced transient amplitude when GABA was applied. Together, these data suggest that calcium transients in immature abGCs are largely GABA dependent.

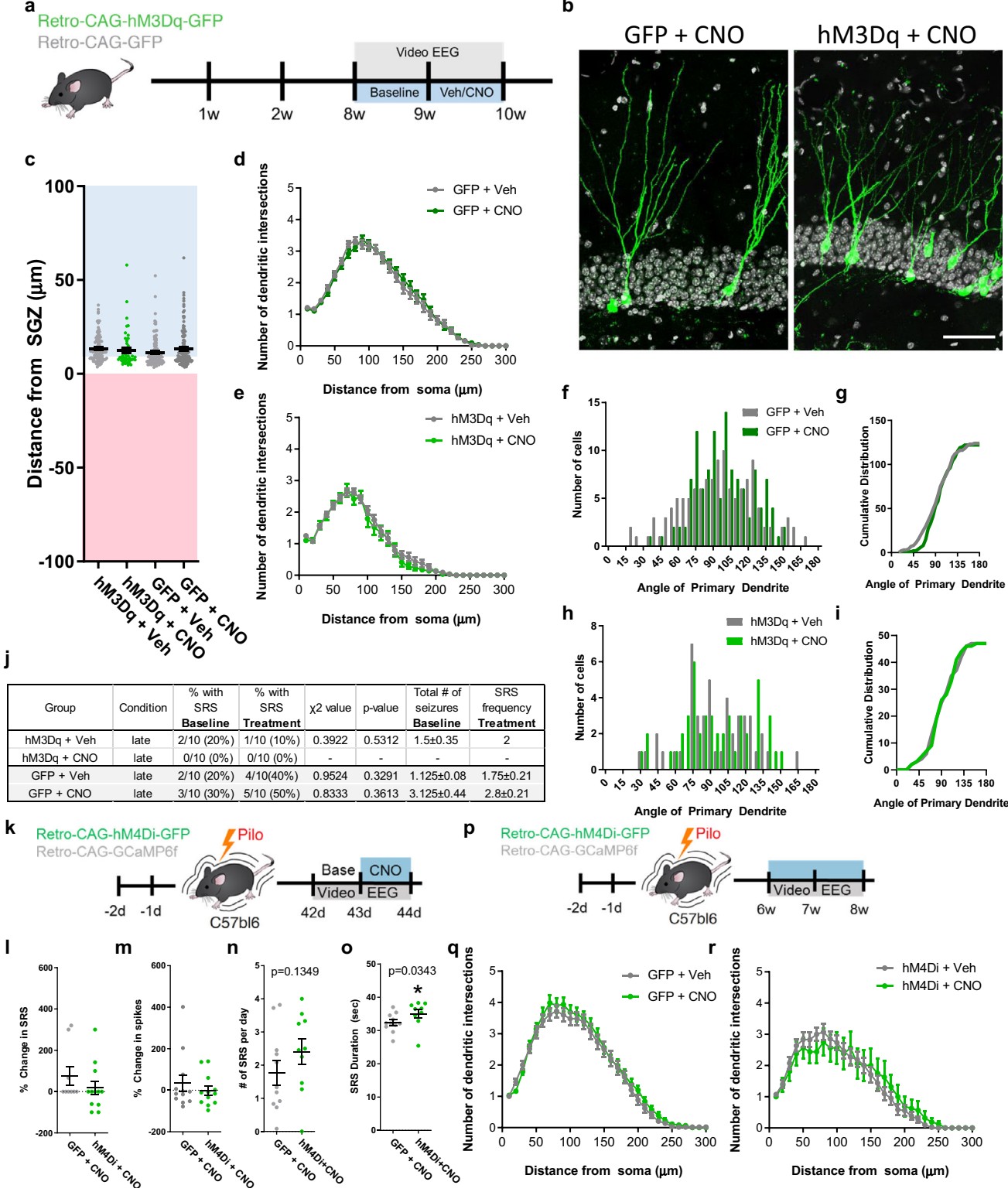

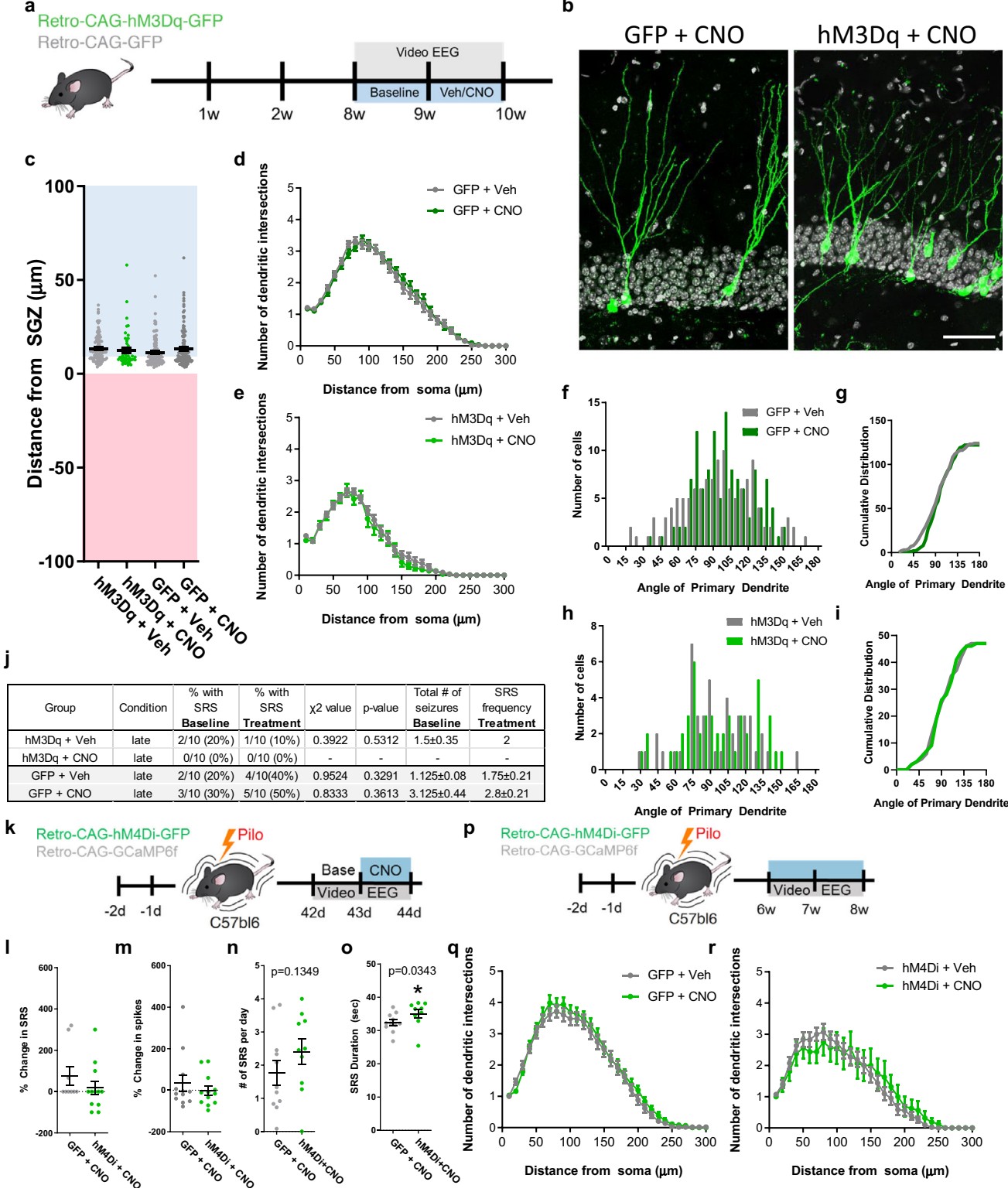

| Group | Condition | % with SRS Baseline | % with SRS Treatment | χ2 value | p-value | Total # of seizures Baseline | SRS frequency Treatment |
|---|---|---|---|---|---|---|---|
| hM3Dq + Veh | late | 2/10 (20%) | 1/10 (10%) | 0.3922 | 0.5312 | 1.5±0.35 | 2 |
| hM3Dq + CNO | late | 0/10 (0%) | 0/10 (0%) | - | - | - | - |
| GFP + Veh | late | 2/10 (20%) | 4/10(40%) | 0.9524 | 0.3291 | 1.125±0.08 | 1.75±0.21 |
| GFP + CNO | late | 3/10 (30%) | 5/10 (50%) | 0.8333 | 0.3613 | 3.125±0.44 | 2.8±0.21 |

To compare intrinsic calcium activity of aberrant abGCs at 2 weeks, mice were injected with retroviral GCaMP6f and SE was induced with pilo. As with the sham mice, brain slices were made and live cell calcium imaging was measured in 2-week-old aberrant abGCs (Fig. 5e). At baseline, we measured an elevated calcium signal compared to immature abGCs in sham (Fig. 5f and Supplementary Movie 4). This was observed by high amplitude peaks with an elevated and sustained baseline, often without any deviation (Fig. 5g). To determine if this enhanced intracellular

calcium was due to increased action potential activity, we applied 1 μM TTX and observed no change in spontaneous event amplitude or frequency (Fig. 5h, i). To determine if these sustained events were also GABA dependent, we applied 20 μM bicuculline and observed a decrease in event frequency, but no change in the amplitude of the few remaining spontaneous events (Fig. 5h, i and Supplementary Movies 5 and 6). Blocking GABA$_A$ receptors had the largest effect on suppressing the enhanced intracellular calcium baseline. Together these results suggest that

**Fig. 3 Activity dependent regulation of aberrant neurogenesis is restricted to an early critical period. a** Mice were injected with retrovirus to express hM3Dq (green) or CAG-GFP (gray) in neural stem cells. Eight weeks following injections, mice were implanted for video-EEG and recorded for seizures. One week of baseline was recorded and then either saline (Veh) or CNO was injected daily. **b** Representative images. **c** Quantification of migration distance. **d** Sholl analysis of dendrite complexity from GFP group (**e**) and hM3Dq group. For **c–e**, $n = 110$ (hM3Dq + Veh:gray), 55 (hM3Dq + CNO;green), 124 (GFP + Veh:green), 143(GFP + CNO:gray) cells per group. Ten mice per group. **f** There was no difference in dendrite angle in GFP. **g** Cumulative distribution of the primary dendrite angle of GFP angle. *$p < 0.01$, Kolmogorov–Smirnov test. **h** There was no difference in dendrite angle in hM3Dq. **i** Cumulative distribution of hM4Di group. **j** Table of spontaneous seizures with each group reporting baseline and percentage of responses to CNO. *$p < 0.05$, Chi-squared test. **k** Experimental design for silencing mature aberrant abGCs. Following retrovirus delivery of DREADDs or control GFP, and pilo-induced SE, EEG recording was performed at day 42 to collect baseline data. At day 43 post SE, CNO was administered and the acute effects were monitored for 24 h. **l** Quantification of the change in SRS from baseline. **m** Quantification of the change in spike activity from baseline. **n** Quantification of the frequency and **o** duration of spontaneous seizures. For **l–r**, $n = 11$ (GFP + CNO), 9 (hM4Di + CNO) mice per group. **p** Experimental design for the chronic effect of silencing aberrant mature neurons. Six weeks following SE, mice were EEG monitored while receiving CNO administration. **q** Silencing mature aberrant abGCs does not alter dendrite morphology in GFP group or **r** hM4Di group. For **q, r**, $n = 58$ (GFP + Veh; gray), 63(GFP + CNO; gray), 44 (hM4Di + CNO;green), 16 (hM4DI + Veh;green) cells per group. *$p < 0.05$, ANOVA with multiple comparisons. All statistics calculated using two-tailed test.

aberrant neurogenesis is characterized by an enhanced baseline of intracellular calcium that is dependent on GABA$_A$ receptor activation. These robust, sustained calcium transients can be altered by GABA during a maturation phase before functional synapses are formed suggesting aberrant abGCs are exposed to elevated GABA following pilo induced SE.

Interestingly, there remains a small component to the spontaneous calcium transients found in immature aberrant abGCs that were not abolished by bicuculline. Our pseudo-typed rabies tracing suggests that aberrant abGCs gain additional glutamatergic synaptic inputs (Fig. 4). While it is unclear from this data if those inputs were established at 2 weeks old, other studies have identified NMDA-mediated synaptic responses as early as in 1-week-old abGCs[35]. To determine if this residual calcium signal is driven by glutamatergic synaptic input, we injected another cohort of mice with GCaMP6f to monitor calcium activity and bath applied the glutamate receptor antagonists, AP5 and DNQX, to block NMDA and AMPA/kainate receptors (Fig. S9A). At baseline, we observed a similar number and amplitude of calcium transient events (Fig. S9B, E, F). With the bath application of glutamatergic antagonists, there was no significant change in calcium transient frequency or amplitude (Fig. S9C, E, F). Interestingly when bicuculline was then applied, most of the events were eliminated, but the same few remaining spontaneous events persisted as seen in Fig. 5h, i. This suggest the remaining calcium transients may be generated from metabotropic glutamatergic synaptic receptors or from intracellular calcium reservoirs.

**DREADDs regulate intracellular calcium in immature abGCs.** To determine if DREADD manipulation can alter intracellular calcium in immature abGCs, we co-injected CAG-hM3Dq-mCherry retrovirus with retroviral GCaMP6f into sham mice and made brain slices for calcium imaging 2 weeks later (Fig. 6a). Cells coexpressing both mCherry and GCaMP6f were imaged and 10 μM CNO was bath applied and responses were recorded (Fig. 6b). At baseline cells displayed calcium transients low in amplitude and frequency (Fig. 6c–f). Upon CNO application, calcium fluorescence showed no change in the frequency of events (Fig. 6d), but an increase in the amplitude of transient events that returned to baseline after washout (Fig. 6e). While there was a trend in the data, there was no significant change in the area under the curve (Fig. 6f). Compared to calcium responses following pilo, hM3Dq-mediated calcium transients were brief and intermittent. Shortly after CNO application, and peak calcium signal was reached, the signal diminished. This suggests that separate mechanisms might increase intracellular calcium after pilo and hM3Dq activation. Next, to determine if the elevated

calcium signal observed in aberrant abGCs can be suppressed using by silencing with hM4Di, we co-injected the retroviral GCaMP6f with a CAG-hM4Di-mCherry retrovirus, injected pilo and similarly made brain slices 2 weeks later for live cell imaging (Fig. 6g). As expected, cells expressing both GCaMP6f and hM4Di-mCherry showed elevated baseline calcium activity (Fig. 6h, i and Supplementary Movie 7). Upon application of CNO, intracellular calcium decreased (Fig. 6j–l and Supplementary Movie 8). During CNO washout, intracellular calcium level increased (Fig. 6j–l and Supplementary Movie 9). Interestingly, some cells returned to a sustained calcium signal, while other cells exhibited a transient dynamic similar to sham baseline (Fig. 6i) or no recovery at all. Additionally, cells expressing GCaMP6f, but not the hM4Di-mCherry, showed no response to CNO application demonstrating CNO had no off-target effects on intracellular calcium activity (Fig. S10). Together, these results suggest that modulation of 2-week-old abGCs with DREADDs alters the regulation of intracellular calcium.

**Discussion.** Our work identifies insights into epileptogenesis in the hippocampus. Pathological changes during epileptogenesis are thought to increase the excitability of the hippocampus circuitry by abnormal abGC migration, formation of hilar basal dendrites, mossy fiber sprouting, interneuron cell loss, and accelerated aberrant neurogenesis[11,14,16,36,37]. While all are associated with epilepsy, there is limited data to demonstrate causality. Further there is no well understood mechanism for the pathogenesis of these cellular phenotypes. We identify a circuit level change attributed to aberrant neurogenesis where excessive activity during a critical period of abGC maturation results in mismigrated abGCs and irregular apical dendritic development, both essential to establish input from the eCTX and appropriate output to the CA3. By silencing aberrant neurogenesis, migration defects are reduced, and dendritic changes are corrected allowing for appropriate cortical inputs to be made. Previous work demonstrates that normal development of abGCs is restricted to a similar critical period of enhanced synaptic plasticity[38,39] and is necessary to maintain the inhibitory-feedback function of the dentate circuitry for information processing[40,41]. Further, GABAergic interneurons guide the maturation and integration of abGCs and are one of the primary cells that die during SE[7,9]. Our data suggests that input to aberrant abGCs from these interneuron populations do not change after SE providing an increased GABA-mediated intracellular calcium signal likely promoting abnormal maturation. Using chemogenetics to mimic these intrinsic cues, our data extends this regulatory mechanism to couple the timing of this activity to mechanisms of aberrant neurogenesis and the capacity to promote pathological circuitry.

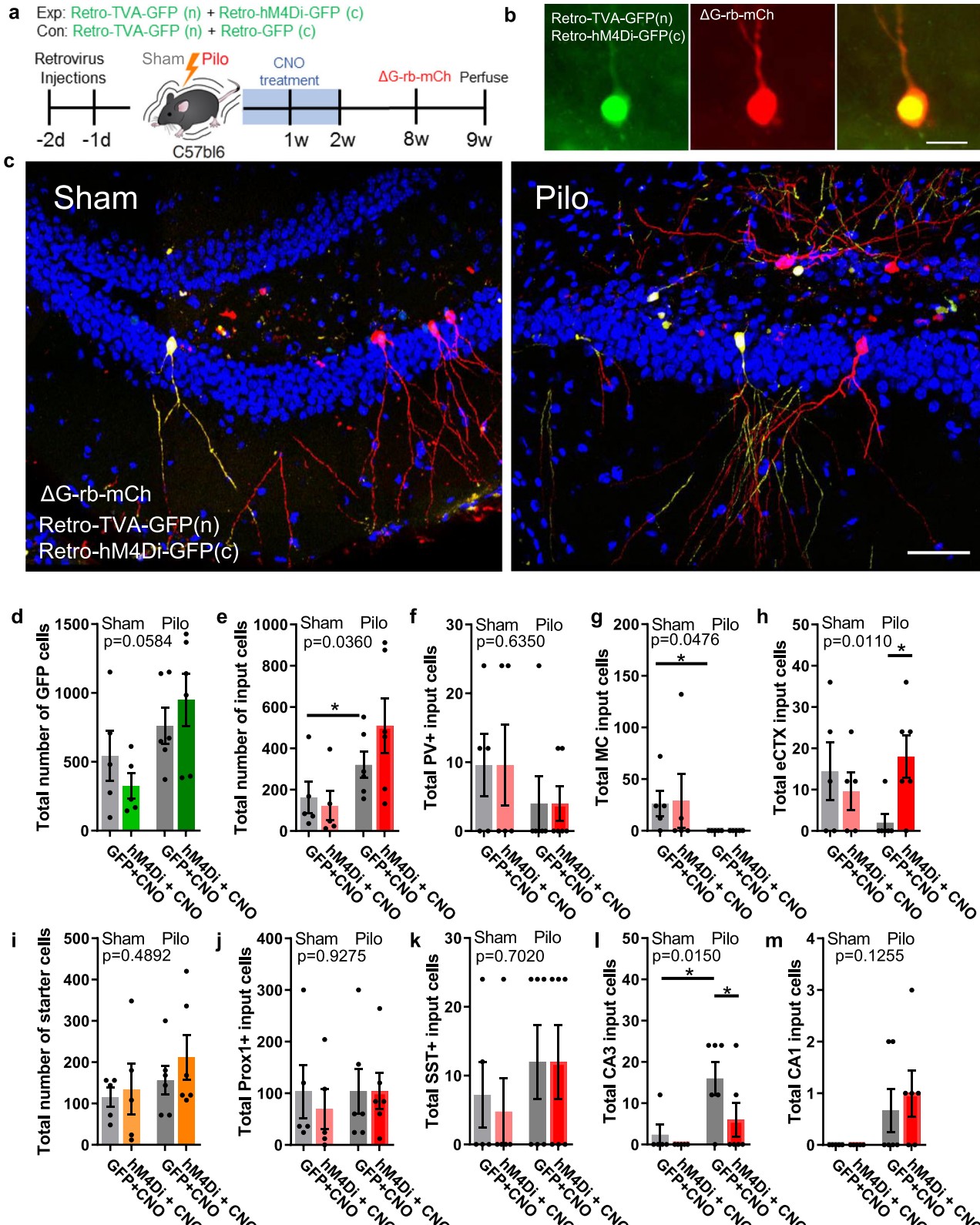

An open question in the field is if hilar EGCs are the source of spontaneous seizures. Previously, we used a transgenic approach to ablate aberrant neurogenesis including EGCs to demonstrate their role in seizures and epilepsy related cognitive deficits[14]. Recent work from others have implicated intrinsic pathways to generate mismigrated abGCs in epilepsy. A conditional knockout of the phosphatase and tensin homolog (PTEN) gene, an upstream regulator of the mTOR pathway, in neural stem cells generates EGCs sufficient to promote SRS[17]. However, conditional knockout of the disabled-1 gene (Dab-1), a regulator of the Reelin pathway, also generates mismigrated abGCs, but fails to promote seizures[42]. However, EGCs compared to normotopic abGCs have a greater excitatory-to-inhibitory input, priming them for increased synaptic drive and increased intrinsic activity[43].

**Fig. 4 Silencing immature adult-born granule cells in pilocarpine model prevents the miswiring of the dentate circuit. a** Experimental timeline of injections. **b** Representative image of a starter cell showing distinct nuclear (n) TVA-GFP and cytoplasmic (c) hM4Di-GFP expression in abGCs with an overlay of the ΔG-rb-mCherry. Scale bar represents 10 μm. **c** Representative images of dentate gyrus with ΔG rabies virus monosynaptic tracing labeling one starter cell (yellow) and input cells (red). Left shows intact sham DG and right is pilo DG with the characteristic abGC sclerosis of the suprapyramidal blade. Scale bar represents 50 μm. Quantification of total number of: **d** cells expressing both nuclear and cytoplasmic GFP (gray bars:GFP + CNO; green bars:hM4Di + CNO), **e** the total number of input cells (mCherry +), **f** PV + input cells (PV +/mCherry +), **g** mossy cell (MC) inputs (GluR2 +/mCherry +), **h** inputs from the entorhinal cortex (eCTX). For **e–h**, gray bars:GFP + CNO; red bars:hM4Di + CNO. **i** The total number of starter cells (GFP +/mCherry +; gray bars:GFP + CNO; orange bars:hM4Di + CNO), **j** inputs from granule cells (Prox1 +/mCherry +), **k** somatostatin (SST) interneuron input cells, **l** input pyramidal cells from the CA3 subfield, and **m** input pyramidal cells from the CA1 subfield. For **j–m**, gray bars:GFP + CNO; red bars:hM4Di + CNO. Data presented as mean ± SEM. $n = 6$ mice per group. *$p < 0.05$, ANOVA with multiple comparisons. All statistics calculated using two-tailed test.

These studies suggest that hilar EGCs may be heterogeneous in their ability to modulate dentate gating and alone are not the likely source of seizure generation. This does not rule out that aberrant abGCs can establish a seizure "hub" network, but that EGCs are not the initiating cell. Our results (Fig. 3k–r) further corroborate that EGCs form seizure hub networks that potentially function to breakdown inhibitory gating mechanisms of the DG. Using DREADDs to manipulate immature abGCs during an early stage of maturation, we alter spontaneous seizures by disrupting the normal development of abGCs. Manipulation of the mature abGCs did not have the same effect on spontaneous seizures supporting that EGCs are not the initiating cell population for seizures. In further support that EGCs form a seizure hub network, we see that by 2 weeks, aberrant abGCs already begin to migrate ectopically and display an abnormal apical dendrite angle. However, at this time point, silencing aberrant neurogenesis did not affect abGC migration or dendritic angle as we saw at later time points, suggesting these phenotypes require more time to develop. Furthermore, this supports that abnormal activity induces the formation of the seizure hub network that perpetuates spontaneous seizures. Thus, the seizure hub network established by aberrant neurogenesis includes a complex, heterogeneous population of abGCs with ectopic migration as a prime cellular component for circuit level disruption.

To better understand how this seizure hub network breaks down the normal gating function of the DG we used rabies retrograde tracing to identify synaptic input changes to the abGCs. We found that input connections from the eCTX could be restored by silencing immature abGCs after pilo. These input connections correlated with the reduction in ectopic abGCs and a restored primary dendrite angle. By preventing ectopic abGC migration and restoring dendrite angle, these cortical input connections are restored. This cortical-to-dentate pathway has been implicated in regulating mood disorders, like depression, a major comorbidity in patients with epilepsy[44,45], and has recently been identified as a bidirectional modulator for DG excitability[46]. In addition, back projections from the CA3 to the DG have been observed in pathological conditions[24]. Our retrograde tracing identified input cells from the CA3 only after pilo, which were reduced by silencing the immature abGCs. Our data suggest these two input populations, eCTX and CA3 back projections, contribute to the generation of a seizure hub. Because the rabies system labeled multiple starter cells, we cannot determine if EGCs in the hilus or the SGZ gained back projections from the CA3 to suggest a possible link between the seizure hub network and comorbidities of epilepsy besides seizures. Interestingly, we observed no change in neighboring Prox1+ GCs, and none of these Prox1+ cells were located ectopically in the hilus (Fig. 4j), suggesting that aberrant abGCs receive a heterogeneous source of recurrent connections, but mismigrated abGCs are not one of them. However, it does not rule out that EGCs are downstream of aberrant abGCs located within the granule cell layer or that they

drive recurrent input onto other GCs outside of the labeled cohort of this experiment. Interestingly, we measured mossy fiber sprouting using the zinc transporter (Znt3) and found when silencing aberrant neurogenesis with hM4Di, we saw no change in Znt3 expression (Fig. S11). Similar to previous studies, this suggest that aberrant abGCs do not contribute significantly to mossy fiber sprouting[14].

Our data suggest that breakdown of the seizure hub network is coupled to a loss of excitatory inputs from the cortex and mossy cells, and a gain of back projections from the CA3 and CA1. However, it is likely that major changes also occur downstream of aberrant abGCs following SE. Interestingly, with both hM3Dq activation (Fig. 1) and silencing with hM4Di (Fig. 2), we measured c-fos activation in the local DG circuit and downstream in the CA3 to determine if output activity was altered. While we observed that the number of seizures correlated with the number of c-fos activated cells, neither hM3Dq nor hM4Di with CNO administration altered the relationship between seizures and activation of c-fos neighboring GCs (Fig. S12). Using c-fos as a read out of neuron activity would suggest spontaneous activity of abGC do not recruit neighboring GCs. However, recent work[47], coupled in vivo two-photon calcium imaging and LFP recordings to measure DG population activity during interictal epileptiform discharges (IEDs), a common electrographic event observed between seizures in epileptic patients. During IEDs, abGCs were recruited to a greater degree than neighboring GCs suggesting abGC play an excitable role in seizure hub networks. While this study did not investigate abGC recruitment during spontaneous seizures, it supports the reorganization of DG networks by abGCs in epilepsy. Further investigation of output anatomy and function of aberrant abGCs is necessary to determine how aberrant neurogenesis reorganizes downstream targets of DG seizure hub networks.

How does hM4Di modulate immature abGC development? Mechanisms for neuronal silencing by hM4Di are mediated through the inhibitory G-protein complex comprised of Gα$_i$ and the tightly associated Gβ and Gγ subunits. Upon receptor activation, the Gβγ subunits dissociate from Gα$_i$ and initiate downstream signaling pathways[48]. There are two notable downstream targets of Gβγ activation, inward rectifying potassium channels (GIRK) and voltage gated calcium channels (VGCCs). However, GIRK channels are not expressed in immature abGCs prior to 3 weeks old. This delayed expression accounts for their more depolarized resting membrane potential, higher input resistance, and increased excitability compared to mature abGCs[49]. Therefore, suppressing intrinsic excitability via GIRK activation is not a likely mechanism for silencing aberrant neurogenesis using hM4Di. An additional target of Gβγ is VGCCs[50,51]. While the role of VGCCs in mature neurons is well understood[52,53], their role in immature abGCs is less defined. Prior to 3 weeks, VGCCs generate low-threshold somatic calcium spikes in immature abGCs[31,33,34].

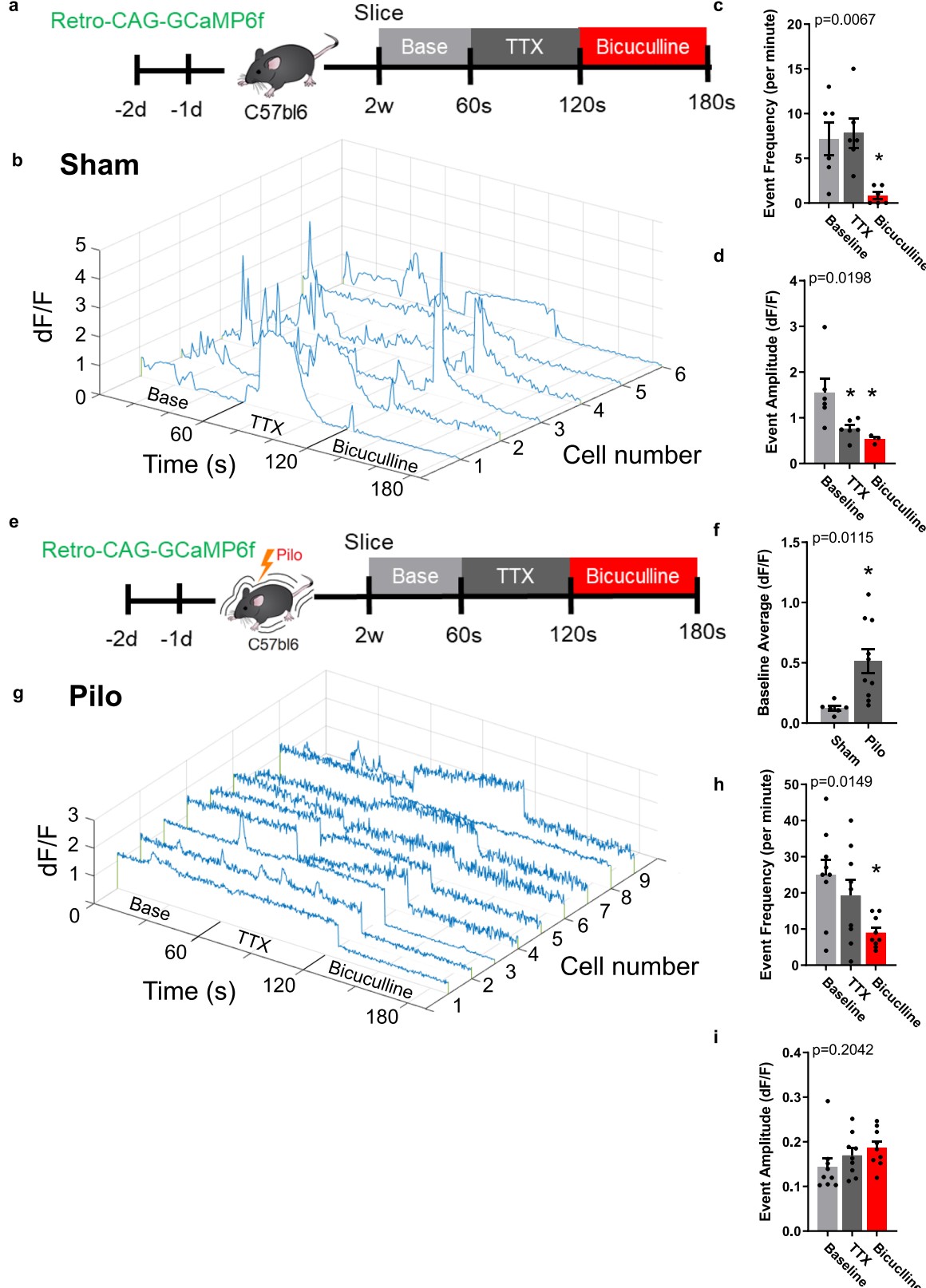

**Fig. 5 Enhanced intracellular calcium in aberrant neurogenesis is regulated by GABA. a** Experimental timeline of calcium imaging in slices from sham mice. **b** Plots of calcium traces from each 2-week-old abGC in the data set. **c** Quantification of calcium event frequency comparing baseline (light gray), 1 μM TTX (dark gray), and 20 μM bicuculline (red). **d** Quantification of event amplitude. **e**. Experimental timeline of calcium imaging in slices from pilo mice. For C and D, $n = 6$ cells from four mice. **f** Quantification of baseline change in fluorescence between sham ($n = 6$ cells) and pilo ($n = 10$ cell) baseline data. **g** Plots of calcium traces from each 2-week-old aberrant abGC in the data set. **h** Quantification of calcium event frequency comparing baseline, 10 μM TTX, and 10 μM bicuculline. **i** Quantification of event amplitude. Data presented as mean ± SEM. For H and I, $n = 9$ cells from three mice. *$p < 0.05$, using ANOVA with repeated measures, except Student's $t$ test for F. All statistics calculated using two-tailed test.

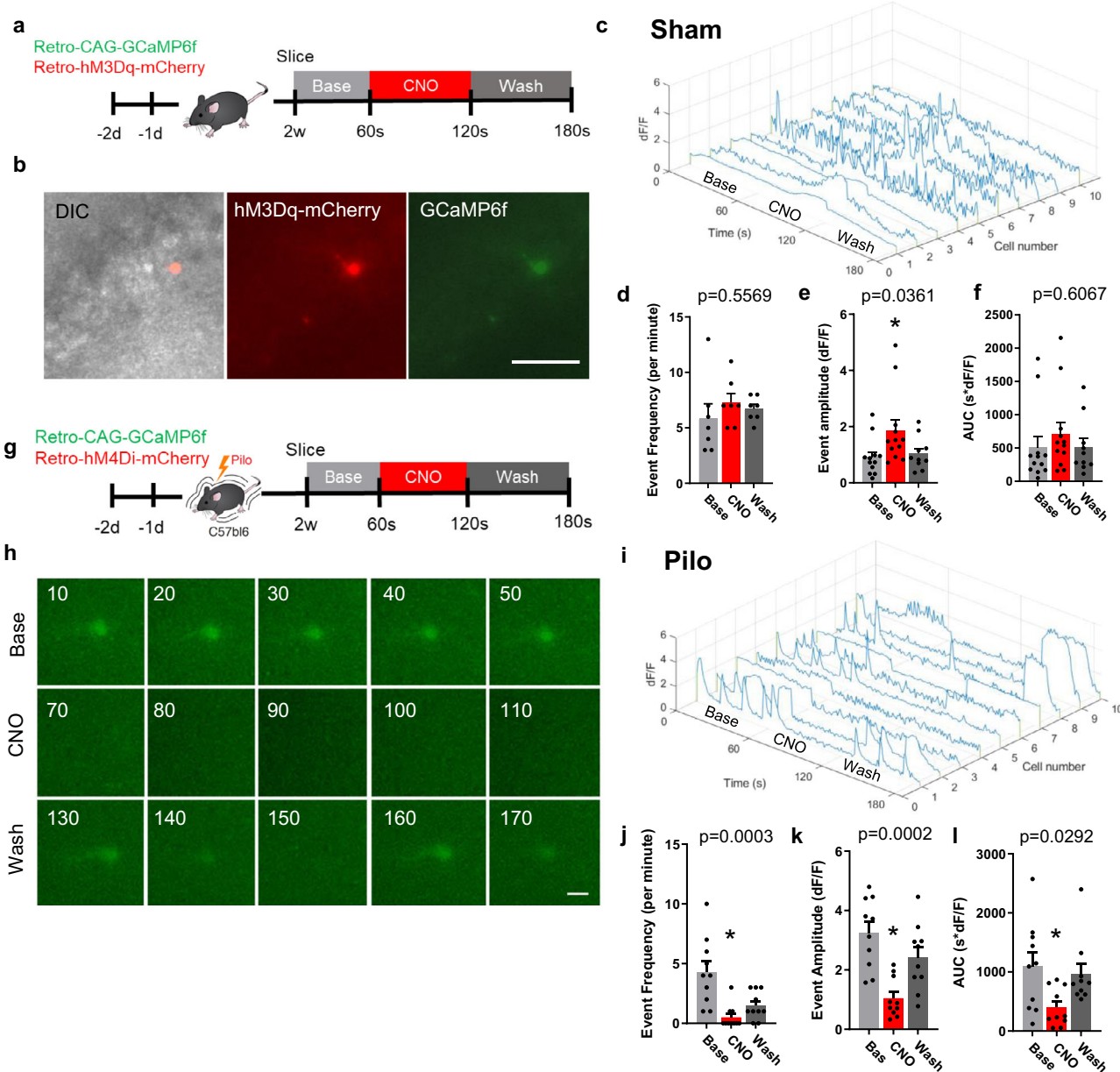

**Fig. 6 DREADDs modulate intracellular calcium of immature adult-born granule cells. a** Sham experimental timeline of injections and slice preparation. **b** Representative images of hM3Dq-mCherry cell co-expressing GCaMP6f. DIC image of abGCs of the dentate gyrus in slice preparation. Scale bar represents 50 μm. **c** Example traces of the change in fluorescence over time from all cells in the dataset ($n = 10$ cells). Baseline (light gray) is displayed over the first 60 s, followed by 60 s of CNO response (red), and 60 s of ACSF washout (dark gray). Quantification of **d** event frequency, **e** event amplitude, **f** area under the curve (AUC). **g** Representative image of hM4Di-mCherry expression in 2-week-old aberrant abGCs. **h** Time lapse representation for 2-week-old abGC. Scale bar represents 10 μm. **i** Example traces of the change in fluorescence over time of all cells in the dataset ($n = 10$ cells). Quantification of **j** event frequency, **k** event amplitude, **l** area under the curve (AUC) for aberrant abGCs. Data presented as mean ± SEM. $n = 4$ mice per Sham group (**d, e**) and 3 mice per Pilo group (**j–l**). *$p < 0.05$, using ANOVA with repeated measures. All statistics calculated using two-tailed test.

Activation of VGCCs in immature neurons are thought to boost suprathreshold action potential generation by supporting membrane depolarization with inward calcium currents and provide trophic cues that promote neuronal development and dendritic outgrowth[54]. Furthermore, RGS-6, the regulator of G-protein 6, plays a functional role in the physiological maturation of abGC through the regulation VGCCs[55]. More studies are needed to determine how hM4Di can alter VGCC function in immature abGCs.

Two-week-old abGCs have a dynamic intracellular calcium response, which can be eliminated by GABA$_A$ receptor antagonist. Immature abGCs express high levels of NKCC1, a cation-Cl$^-$ importer, which changes the reversal potential for Cl$^-$, causing GABA$_A$ receptor activation to depolarize the cell[7]. This GABA-mediated depolarization activates low-threshold VGCCs to increase intracellular concentrations of calcium[34]. In the DG, the primary sources of GABA are inhibitory interneurons, PV and SST. Following SE, these populations are either over stimulated and die via excitotoxicity[19], or survive and become persistently hyperactive potentially dumping excess GABA into the hilar niche[20]. Results from our monosynaptic rabies experiments suggest interneuron inputs (PV and SST) to aberrant abGCs do not drastically change in pilo treated mice (Fig. 4f, k). However, compared to sham, 2-week-old aberrant abGCs

from pilo treated mice, had an amplified baseline of GABA$_A$ receptor-mediated calcium activity. Upon silencing with hM4Di, this intracellular calcium amplification was eliminated. The increased calcium signal in aberrant abGCs offers a mechanism for abnormal regulation of gene expression programs in epileptogenesis. While it is unknown what sustained elevation of intrinsic calcium levels has on abGC development and maturation, the role of calcium in neurodevelopment has been widely studied. In mouse cerebellar GCs, the amplitude and frequency of calcium transients, via VGCCs, dictate the rate of neuronal migration, suggesting calcium acts as a speedometer to integrate various intrinsic/extrinsic cues that drives neuronal migration[56]. This can occur through calmodulin, a calcium binding protein, that interacts with a complex network of signaling molecules including calcium/calmodulin dependent kinase II (CaMKII)[57], Ras/Raf/MEK/ERK pathway[58], and ultimately gene expression programs. Calcium represents a convergent signal where multiple modalities have the potential to regulate the process of abGC maturation. Our results which demonstrate that hM4Di regulate intracellular calcium in aberrant abGCs offers a new way to define "silencing" a neuron. Counter to the classical definition of synaptic function, silencing intracellular calcium, especially under pathological conditions expands the definition beyond suppressing membrane properties to silencing potential aberrant gene regulatory pathways.

Aberrant neurogenesis highlights the negative impact a limited number of abGCs can have on overall brain function. This study provides mechanistic insight into aberrant neurogenesis, a pathological hallmark of TLE. We identified a critical period for aberrant neurogenesis where sustained and robust calcium transients are associated with abnormal maturation of abGCs that can establish an epileptic hub network to promote spontaneous seizures. While this hub alone does not drive seizures, selectively suppressing this enhanced calcium during this critical period can prevent the hub from being established resulting in reduced spontaneous seizures. Downstream regulators of this aberrant calcium signal in immature abGCs are potential targets for anti-epileptic drugs.

## Methods

**Animals**. All mice (C57bl/6, all female) were purchased from Envigo at 5–6 weeks of age and housed in the animal facility with 12-hour-light/dark cycle, room temperature 65–75 °F(18–23 °C) with 40–60% humidity. Mice were given access to standard chow and water ad libitum. All mice were group housed (five per cage) except during video-EEG monitoring where they were single housed. All experiments were approved and performed in compliance with the animal care guidelines issued by the National Institutes of Health and The University of Texas at San Antonio Institutional Animal Care and Use Committee (approved protocol no. MU112).

**Retrovirus preparation**. All retroviruses were packaged by Lipofectamine transfection (Lipofectamine 2000, Invitrogen) of 293 T cells with 7.5 μg of viral vector, helper plasmids 5 μg CMV-GagPol, and 2.5 μg CMV-VSVG. Forty-eight hours after transfection, the media containing virus was collected and replaced for 3 days. The media was purified (ViraTrap, Biomiga, V1172-01) and concentrated via ultracentrifugation. Purified virus was estimated around 2.6 x 10 e7 i.u. ml⁻¹. Retro-hM3Dq/hM4Di constructs were obtained from A. Schinder (Leloir Institute, Buenos Aires, Argentina) and cloned into the retrovirus backbone pUX-p2A-EGFP (from S. Ge) using PCR based cloning with the following primers: hM3Dq forward ggatcctgaccttgcacaataacagtacaacct, hM3Dq reverse gaatcctcaaggcctgctcgggtg, hM4Di forward ggatccatggccaacttcacacctgtcaa, hM4Di reverse gaatcccgatatcgcggccgccta. For retro-hM3Dq-mCherry and retro-hM4Di-mCherry, hM3Dq/hM4Di were cloned into the retro-CAG-IRES-mCherry retrovirus backbone using PCR based cloning with the following primers: hM3Dq forward cgcggatcctgaccttgcacaataacagtacaacct, hM3Dq reverse gggaccggttcaaggcctgctcgggtg and hM4Di forward cgcggatccatggc caacttcacacctgtcaa, hM4Di reverse gggaccggtcgatatcgcggccgccta. The retro-Syn-TVA-GFP expresses a nuclear GFP, avian viral receptor TVA, and rabies glycoprotein under the synapsin promoter was obtained from H. van Praag (NIA NIH, Maryland)[25]. For calcium imaging, the retrovirus retro-CAG-GCaMP6f was also from S. Ge.

**Rabies virus preparation**. G-deleted mCherry rabies virus vector (Δg-mCherry) and helper lines, B7GG and BHK-EnvA cells, were obtained from Salk GT3 Core (La Jolla, CA). Production and pseudotyping of rabies virus was according to previous methods[59]. Briefly, un-pseudotyped Δg-mCherry virus was amplified by infecting B7GG cells and collecting media containing un-pseudotyped virus and replacing with fresh media after 4 days. Virus was collected again 2 days later. Pseudotyped rabies virus (Rb-EnvA-Δg-mCherry) was made by infecting BHK-EnvA cells with the collected unpseudotyped media, then concentrated with ultracentrifuge. Titer for rabies virus was estimated to be ~1.5 x 10 e7 i.u. ml⁻¹. Infection specificity was determined by infecting EnvA- Δg-mCherry with control CAG-GFP (Fig. S5).

**Stereotaxic surgery/virus injection**. Mice were anesthetized using isoflurane (2–3%) at 1 L/min in a 70% nitrous oxide and 30% oxygen mixture. High titer and purified murine oncoretroviruses were injected into the dorsal and ventral hippocampus to express genes of interest to actively proliferating cells at stereotaxic coordinates from bregma: AP1 −2.0 mm, ML1 ± 1.5 mm, DV1 −2.0 mm; AP2 −2.75, ML2 ± 2.8, DV2 −2.5. In each site, 1 μl of virus was injected over the period of 1 min using a 10 μL Hamilton syringe. Twenty-four hour later each mouse received a second retrovirus injection to maximize the number of neural stem cells infected. For rabies virus injections, Rb-EnvA-Δg-mCherry was injected 8 weeks after the retro-Syn-TVA-GFP was injected into the same coordinates and transcardially perfused with 4% paraformaldehyde 7 days later.

**Epilepsy model**. Following 2 consecutive days of retro-hM4Di-GFP injections, we performed the pilo induced SE model for TLE based on the previously published methods[14]. Briefly, scopolamine methyl nitrate at 2 mg/kg (Sigma-Aldrich S2250) and terbutaline hemisulfate salt at 2 mg/kg (Sigma-Aldrich T2528) were injected intraperitoneally (i.p.) to aid in respiration and peripheral effects due to pilo. 30 min later, pilo hydrochloride (Sigma-Aldrich P6503) was injected i.p. at 300 mg/kg. Mice were then placed in an incubated chamber (ThermoCare) at 31 ºC until SE was reached (about 20 min). Once in SE, mice were transferred to a room temperature cage for 3 h where SE was terminated with diazepam (10 mg/kg; Sigma-Aldrich D0899; i.p.). Mice were given 1 ml of 5% dextrose solution (i.p.) and 1 mL of 0.9% NaCl saline (i.p.) to aid in recovery. Mice were monitored in the incubated chamber for 2 days then returned to their home cage and group housed. Mice were excluded from further experiments if SE was not reached within 1 hr of pilo administration and a full duration of 3 h. For mice injected with retro-hM3Dq or hM4Di, CNO (Sigma-Aldrich C0832) was injected (i.p.) daily or twice daily as noted per experiment at 1 mg/kg. To account for any off-target effects of CNO[60], appropriate controls were included in all analysis.

**Video-EEG recording and analysis**. Spontaneous seizure activity was monitored by 24 h continuous video-EEG recordings. Seven weeks following pilo induced SE, mice were implanted with a subcutaneous radio transmitter (TA11ETAF10; Data Sciences International) with leads connected to stainless steel screws (no. 00-96×1-16; PlasticsOne) implanted in the scull at coordinates from bregma: positive lead AP-2.0 mm, lateral 2.2 mm; negative lead AP 1.0 mm, lateral 2.0 mm. Video-EEG data was recorded for 2 weeks and analyzed using Penomah and Neuroscore software (Version 3.0; Data Sciences International) by a user blinded to experimental groups. Convulsive seizures were identified by a burst of spiking activity (>3 Hz) that persisted for >10 s with spike amplitudes greater than 2x background amplitude. All convulsive seizures were confirmed by video.

**Immunohistochemistry**. Mice were anesthetized and transcardially perfused with cold PFA in 0.1M PBS. Brains were dissected and postfixed in 4% PFA overnight and cryoprotected in 30% sucrose in 0.1M PBS. Forty micrometer coronal sections were cut on a freezing microtome. Immunohistochemistry was performed by free-floating tissue. For chicken anti-GFP, Tyramide-Plus signal (1:50, PerkinElmer NEL70100KT) amplification was performed by removing endogenous peroxidase activity (0.3% H2O2, 30 min at room temperature). 3% normal donkey serum and 0.3% Triton-X-100 in TBS was used to block any nonspecific binding. Primary antibodies used: chicken anti-GFP (1:500, Aves Labs GFP-1020), rabbit-anti-Prox1 (1:500, Millipore AB5475), rat anti-somatostatin (SST; 1:500, Millipore MAB 354), mouse anti-parvalbumin (PV; 1:500, Millipore MAB1572), rabbit anti-GluR2 (1:500; Millipore Sigma AB1768-I), mouse anti-c-Fos (1:500; Santa Cruz sc-8047), rabbit anti-ZNT3 (1:500; Millipore Sigma ABN994). For double labeling, primary antibodies were separately labeled with GFP amplification first (e.g., GFP*/Prox1, GFP*/PV, GFP*/SST). Secondary antibodies used: biotin anti-chicken (1:200, Jackson ImmunoResearch), CY5 secondary (1:200; Jackson ImmunoResearch). Sections were counterstained with DAPI (4′,6-diamindino-2-phenylindole; 1:5000, Roche 236276) and mounted on static charged slides with Polyvinyl alcohol DABCO medium (Sigma-Aldrich 10981).

**Microscopy and morphology analysis**. Quantification of cell numbers were performed by someone blinded to experimental group on a confocal microscope (LSM700; Carl Zeiss Microscope and DMi8 SPE, Leica). To confirm double positive cells Z planes were scanned to determine co-localization. Every 12th section was analyzed, and the number of cells counted were multiplied by 12 to

estimate the total number of cells per animal. Dendritic analysis, migration distance, and spine quantification were performed using stereology quantification on an Olympus BX51 Microscope, MicroFIRE A/R camera (Optronics, Goleta, CA), Optical Fractionator Probe with Neurolucida software (MBF Bioscience, Micro-BrightField, Inc., VT). For migration parameters, distance was measured from the center of the cell to the base of the SGZ. For primary dendrite angle, all angles were medially referenced. Only cells with a cell body and full dendritic tree were included in analysis. Analysis of c-fos positive cells was performed using ImageJ software (NIH). Internal positive control pixel intensity was determined from c-fos positive cells in the cortex of each section. Cells of equal or greater pixel intensity were included in the dataset. For Znt-3 analysis, area of staining was measured using ImageJ software (NIH). Briefly, the entire pixel area of the DG was measured. Using built in threshold tools, the pixel area of ZNT3 staining was measured and subtracted from the total dentate area. This percentage is represented in the data as %Znt3 staining of the DG.

**Slice preparation and calcium imaging**. Ten to 14 dpi after stereotaxic virus injections, mice were deeply anesthetized with isoflurane. In the absence of tail and toe pinch responses, mice were rapidly decapitated and brains were removed and placed in partially frozen modified artificial cerebrospinal fluid (ACSF) containing (in mM): 110 choline chloride, 2.5 KCL, 1.3 NaH$_2$PO$_4$, 26 NaHCO3, 20 Dextrose, 0.5 CaCl2, 7 MgCl$_2$, 1 Kynurenic Acid, 1.3 Na-ascorbate, and 0.6 Na-pyruvate. Horizontal brain slices containing the hippocampus were cut at 250 um thickness using a vibratome. Slices were transferred to warm (34 °C) physiological ACSF containing (in mM): 126 NaCl, 2.5 KCl, 1.25 NaH$_2$PO$_4$, 10 Dextrose, 2 CaCl$_2$, 1 MgSO$_4$, 26 NaHCO$_3$, 1.3 Na-ascorbate, and 0.6 Na-pyruvate. After 30 min, slices were maintained in physiological ACSF at room temperature until imaging.

For imaging, slices were transferred to transparent membranes (30 mm, 0.4 μm, Millipore PIC03050) in fresh ACSF and imaged immediately. Imaging was performed at room temperature on a Leica DMi8 WF microscope with a Leica DFC7000T camera. Images were captured at 2 frames per s with an exposure of 100 ms for 180 s to reduce photobleaching. A baseline measurement was recorded for the first 60 s followed by bath application of drug (1 μM TTX, 10 μM bicuculline, 10 μM CNO, 10 μM AP5, 10 μM DNQX; made fresh daily from frozen stocks). For washout, fresh room temperature ACSF was added and response was recorded for 60 s. Cells were identified by location in the DG and with the morphology of a single primary dendrite. All cells included were located in the GC layer. All cells exhibiting baseline transients were included in the dataset. For pilo treated group, cells were considered dead and excluded if they showed no response to bicuculline.

Image analysis was performed with LAS X software (Leica). Fluorescence as a function of time was measured using onboard tools by selecting regions of interest (ROI) around the soma and a local background ROI. dF/F was calculated as (F(t) − Fbase)/(Fbase − background). Fbase was calculated by generating an all-data histograms from baseline datasets using the largest peak amplitude as Fbase. For event detection, peaks were identified using the PeakCaller, a MATLAB script designed by the Hussman Institute for Autism[61]. Peaks with a 10% rise over baseline were marked. Any peak below 2 standard deviations from baseline were excluded from the final quantification. For area under the curve (AUC), dF/F for each cell was exported and analysis using standard toolboxes in MATLAB (R2017b).

**Statistics and Reproducibility**. For all data, mean is presented as mean ± SEM. Data were tested for normality using Shaprio–Wilk tests or the D'Agostino–Pearson omnibus tests. Statistical differences comparing means were analyzed using a two-tailed Student's t test or ANOVA for data with equal variances. For continuous data sets, Kolmogorow–Smirnov test were used to compare cumulative distributions. Pearson's chi-squared test ($χ^2$) was used for all categorical data sets. For calcium imaging, one-way ANOVA with repeated measures was used and p values are reported for all datasets. Values of $p < 0.05$ were considered significant. All statistics was performed using GraphPad with Prism 8.4.3, following its Statistics Guide.

For Figs. 1–4 and supplementary figs. 1–7 and 11, 12, experiments were repeated three times and the final datasets were analyzed together. The data from Figs. 5–6 and supplementary figs. 8–10 were repeated twice. The final data sets were combined and analyzed together.

**Reporting Summary**. Further information on research design is available in the Nature Research Reporting Summary linked to this article.

## Data availability
All data supporting the findings of this study are provided within the paper and its supplementary information. A source data file is provided with this paper. All additional information will be made available upon reasonable request to the authors. Source data are provided with this paper.

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

## Acknowledgements

We thank Jose Cabrera for support with graphics and Aline McKenzie for manuscript editing, Mary-Colette Lybrand for support and feedback, Juan Song and Alejandro Schinder for DREADD retroviral constructs. This work was supported by grants from NIH (R01NS093992, R01NS113516, and R21AG066496 to J.H.) and (R01NS089770 and R21NS090926 to J.H. and S.G.), Department of Defense (W81XWH-15-1-0399 to J.H.), Semmes Foundation, Inc. (to J.H.), Robert J. Kleberg, Jr. and Helen C. Kleberg Foundation (to J.H.), and American Epilepsy Society Postdoctoral Fellowship (to Z.R.L.).

## Author contributions

Z.R.L. contributed to the concept, design, experimentation, analysis of data, and wrote the manuscript; S.G. contributed to experimental design, analysis, and technical expertise for slice physiology; J.Z., C.S., N.M., and V.J. collected data for confocal counting; M.A. collected data for cell morphology data; L.Z. packaged virus and analyzed EEG data; P.V. helped with validating immunohistochemistry and contributed to conceptualization of experimental design; K.O.C. contributed to conceptualization and design of experiments; S.G. contributed to the experimental design, manuscript review, and provided resources and financial support; J.H. helped design and conceptualize experiments, helped write and edit manuscript, and provided financial support.

## Competing interests

The authors declare no competing interests.
