## [Peer Review File · Nature Communications]

Reviewers' Comments:

Reviewer #1:

Remarks to the Author:

In a previous paper Dr. Hsieh and colleagues showed that pilocarpine-induced epilepsy in mice is associated with newborn granule cells migrating to the hilus (eGCs) instead of migrating into the GC layer (Cho et al. 2015, Nat Comm 6:6606). Furthermore, in the previous paper it was shown that ablating adult neurogenesis using the TK-GCV mouse model, modulated the development of pilocarpine-induced seizures. Remarkably, while single injection of GCV reduced the seizure frequency (SF) by about 30% compared to pilocarpine-only, the seizures were not affected by two subsequent GCV injections. Furthermore, seizure duration was not affected by any GCV treatment. In the new paper 'A critical period for aberrant neurogenesis rewires hippocampus circuitry to cause epilepsy', the authors try to investigate how the newborn granule cells might modulate seizure frequency in the pilocarpine mice. They use DREADD receptor mediated silencing of a cohort of newborn granule cells either 1-2 weeks or 9 weeks after pilocarpine injection. The authors conclude, that 'silencing of aberrant adult born GCs in a temporal lobe epilepsy model prevented abnormal development and SRS (spontaneous recurrent seizures), thereby restoring the protective function back to the dentate gyrus'. Unfortunately, neither were SRS 'prevented', nor was any protective function 'restored' to normal. Most importantly, I cannot see a 'circuit level mechanism' as claimed by the authors.

1. Silencing of 2 wpi and 9 wpi GCs. The authors expressed DREADD receptors in a cohort of GCs born 1-2 days before pilocarpine injection. These adult born granule cells (abGCs) were silenced at different time points afterwards. Fig2. shows that silencing a cohort the abGC directly after pilocarpine injection for 2 weeks, reduces SRS frequency to about 40%, while the SRS duration is unaffected. By contrast silencing the abGCs 9 weeks after pilocarpine (Fig. 3) does not affect seizures at all. This is also consistent with the previous observation by the same authors, showing that complete ablation of adult neurogenesis does not affect seizure generation (Cho et al. 2015, Fig. 6). This indicates that abGCs may under some conditions potentially affect downstream hippocampal neurons generating the seizures, but are not themselves included in seizure generation. To understand the mechanisms underlying this phenomenon it would be important to study postsynaptic targets of the abGCs and how hyperactive abGCs can change downstream circuits. However, this was not investigated. As a first step, one could measure cFOS expression in downstream cells +/- CNO-mediated silencing or after Chr2-mediated stimulation of abGCs.

2. Activating abGCs. Conversely, Fig. 1 shows that CNO-mediated activation of a cohort of abGC for about 1 week can trigger the development of spontaneous seizures. However, this EEG activity is much shorter (13s, Fig. 1K) than after pilocarpine (30s, Fig 2LM). Also for 0-1w CNO there is no duration given in the table – why? How do we know that the CNO-induced activity is the same as the pilocarpine-induced activity? Most importantly, the downstream synaptic modifications of CNO activation of abGC was not investigated. How do the mossy fibers looks like? Does the size or the number of MFBs increase? What about filopodial extensions in CA3?

3. Dentate function. On a similar note, the authors claim that the dentate constitutes an inhibitory feedback circuit. However, the recurrent feedback inhibition does not exist in 1-2 week old cells, but develops well after 4 weeks post mitosis (Temprana et al. 2015). By contrast feedforward excitation and inhibition in CA3 develops much earlier and could contribute to the effects observed by Hsieh and colleagues. Only considering feedback inhibition is too simplistic.

4. Upstream connectivity. Fig. 3 shows some synaptic inputs to abGCs after pilocarpine induced seizures or after CNO treatment. How does this compare with healthy controls? Why we don't see mossy cell inputs to abGCs? Fig. 4a need to be shown at much higher resolution. Nothing can be seen at present.

5. Pilocarpine model. The pilocarpine model is special. How do the present results relate to other

epilepsy models as for example the kainite model? The authors imply that pilocarpine is representative for all TLE.

Minor

1. How does the distribution of dendrites in Fig 2 JK compare to control? What about quantitative analysis fitting a gaussian distribution? Skewness of distribution in Fig. 2G could be quantified with fitting a skewed function to the distribution.
2. What does the table in Fig. 3j represent? No comment in legend.
3. Fig. 4 panel labeling does not correspond to legend.

Reviewer #2:

Remarks to the Author:

In the manuscript "A critical period for aberrant neurogenesis rewires hippocampus circuitry to cause epilepsy", Lybrand et al described the critical period impacting on the morphology of newborn neurons after seizure. Seizure induces aberrant neurogenesis in the dentate gyrus and newborn neurons generated after seizure display abnormal morphology and ectopic localization in the DGs - factors implicated in the development of spontaneous seizure activity. The authors present data suggesting that early neuronal activity of newborn neurons influences their migration, final morphology and the neuronal circuit. Increased neuronal activity after seizure leads to abnormal morphology of newborn neurons and resulted in the development of spontaneous seizure. Using DREADD and rabies virus-based neuronal circuit tracing, the authors demonstrated that silencing early neuronal activity of newborn neurons restores their proper morphology and connections to the hippocampal circuit.

The idea that silencing early neuronal activity after seizure is beneficial for restoring proper morphology and neuronal circuit function is interesting. The methods used to support their findings are reasonable and convincing. I am concerned though that they haven't provided any clear idea about how early neuronal activity of newborn neurons might directly regulate morphology. The authors demonstrate the outcome of silencing the neuronal activity after seizure but provide little more insight. They did not give any data to explain how/why early neuronal activity affects the establishment morphology or aberrant positioning. Below are some suggestions about ways to improve the study and make it more authoritative

1) The authors indicated that silencing early neuronal activity of newborn neurons restores proper morphology. This suggests that immature newborn neurons receive neuronal activity from surrounding neurons, affecting morphology when they become mature neurons. Presumably, this isn't only true after seizures but it seems rational to assume that newborn neurons produced in physiological conditions also receive neuronal activity from surrounding neurons. Thus, it seems reasonable to examine the differences in the neuronal activity of newborn neurons produced in physiological conditions vs after seizures and to determine whether newborn neurons produced after seizures receive excessive neuronal activity / inputs than controls. From their data and manuscript, I interpreted that newborn neurons receive excessive neuronal activity after seizure, thus they tried to repress it. Also Line 68 says "These overexcited interneurons thus impose a hyperexcitable signal to mature new neurons". However, unless I am missing the critical reference there isn't a clear message in the literature that immature neurons receive excessive neuronal activity from GABAergic neurons or other local neurons within 2 weeks from the born. Perhaps the authors can show some evidence that newborn neurons after seizure have increased neuronal activity / more inputs from surrounding neurons in the critical period, by either citing the relevant papers or adding some additional experiments.

2) It would be useful to more clearly associate the abnormal morphology of immature neurons and the influences on their connectivity during early stages. The authors showed that newborn neurons produced after seizure have abnormal morphology when they become mature. However, the authors did not show whether this morphological abnormality is already detectable from early stages (during the 2 weeks after seizure). This would help to establish whether the abnormal morphology is induced by abnormal activity or whether the activity induces the morphology.

3) Relate to the question 2, they showed that early neuronal activity of newborn neurons affects their morphology. However, they only showed the data at 8/9 weeks after CNO injection, and there is no data about the morphology of newborn neurons before they become mature. About this experiment, I am wondering if they can also address if morphological differences are observed immediately after 2 weeks treatment of CNO.

4) In figure 4, they injected two retroviruses into the DGs (one expresses TVA, the other is for hM4Di). However, both viruses express GFP. Thus, using only GFP expression, the author cannot distinguish if the cell is carrying both TVA and hM4Di. If they used only GFP to detect starter cells in the hM4Di group, their analysis would contain the cells that carry only TVA, but not hM4Di. I am wondering if they could show how many GFP+ cells (what percent of GFP+ cells) in hM4Di group have both TVA and hM4Di.

5) The authors showed that silencing early neuronal activity restores proper neuronal circuitry in seizure-induced newborn neurons. However, they did not show the neuronal circuit of newborn neurons in the physiological condition. This makes it difficult to interpret if the silencing early neuronal activity really contributes to restoring the proper neuronal circuit or yet a distinct type of circuit. If they say "restore", it would be helpful to show the neuronal circuit of adult-born neurons in physiological conditions with/without silencing of neuronal activity.

6) The authors proposed a critical period in which increased neuronal activity influences the morphology of newborn neurons, resulting in abnormal neuronal circuit. In figure 4, they identified the neuronal circuit of these seizure-induced neurons. However, given that silencing early neuronal activity can restore the proper neuronal circuit, there must be some local neurons projecting to immature newborn neurons during the critical period, and these neurons are responsible for the increasing neuronal activity and changing the morphology of newborn neurons after seizure. They assumed GABAergic interneurons give a tonic signal during the critical period. If so, I am wondering if they could perform similar rabies virus experiments during 2 weeks after pilocarpine treatment to identify the neurons projecting to the immature newborn neurons, and also if they could ask whether the subtype of these projecting neurons are different between the physiological conditions and after seizures.

Reviewer #3:

Remarks to the Author:

The authors of this manuscript identify a developmental time window when excessive activation of adult-born dentate neurons alters hippocampal circuitry and drives spontaneous seizure activity. They show that activation of 0-2 week-old adult-born cells with DREADDs is sufficient to generate ectopic migration and spontaneous seizure activity (Fig 1), and that suppressing adult born cells with inhibitory DREADDs during the same period after pilocarpine-induced seizures reduces the number of ectopic GCs and spontaneous seizure frequency (Fig 2). The effects of activating and inhibiting older adult-born neurons did not replicate the effects on 0-2 week old cells, indicating the presence of a critical-period (Fig 3). Interestingly, silencing adult-born cells during the critical period prevented abnormal connectivity in the EC-DG-Ca3 circuit (Fig 4).

The main novel aspects of this work are that activation of adult-born cells with DREADDs is sufficient to trigger EGCs and generate spontaneous seizures (Fig 1), and the results from Fig 4

showing altered wiring associated with ectopic migration and rescue. Evidence that new neurons are sensitive to seizure-induced aberrant development only during a restricted period of maturation corroborates prior work (i.e. Kron et al., 2010, among others). Together these findings provide novel insight into the pathological capacity of adult-born neurons in disease states.

Overall the experiments are well-designed and appear to be rigorously performed and analyzed. Most of the conclusions are well-supported, although I think that one issue needs further consideration.

Major

Line 54. The authors conclude that "excessive activity early in GC maturation generates aberrant neurons that have abnormal migration....." and they use the terms activate and suppress to describe the DREADD manipulation. Excitatory and inhibitory DREADDs are typically used to increase and decrease spiking in neurons, so it is unclear they affect cells that may not fire action potentials (i.e. progenitors). In order to understand the mechanism by which adult-born GCs undergo abnormal maturation and the extent to which chemogenetics mimic extrinsic cues during epileptogenesis, the authors should determine how their CNO "activates" or "suppresses" retroviral-labeled cells. These manipulations may alter intrinsic excitable properties, but an alternative possibility is that they act via intracellular signaling pathways that are independent of changes in membrane properties and thus do not "activate" or "suppress" in a manner that is analogous to activation or suppression of spiking activity in neurons.

A related point is that there is no justification for the duration of the CNO treatment regime – was one week necessary to see an effect? If chemogenetic activation of new cells mimics extrinsic cues that drive abnormal migration, connectivity and SRS after SE, then the pathological changes would likely be triggered by a shorter treatment of CNO that lasts the duration of status epilepticus.

Minor issues:

Figure 1K. The seizure duration seems to be absent from the 0-1w group in the Table.

At several places in the manuscript the authors focus on the DG as an inhibitory feedback circuit and the putative role of adult-born neurons in feedback inhibition. However, this focus on feedback inhibition is not supported by citations, and is a little distracting since there are several forms of inhibition that maintain gating function. More importantly, very recent results seem to rule out a potential role of newborn neurons in conventional feedback inhibition mediated by GABAA receptors (in favor of an unconventional form of direct inhibition, Luna et al., 2019). Since the current results don't address the role of newborn cells in inhibitory function, the authors might want to tone down this focus.

Line 66. Presumably it is meant that PV and SST are over-excited before they die. Regardless, there is conflicting data about what happens to particular interneuron subtypes in mouse epilepsy models (i.e. Huusko et al., 2015; Buckmaster et al., 2017).

Line 82. Since no dendrite abnormalities were detected, this text is confusing, "To determine if abnormal migration and dendrite changes..."

Line 125. Loss of hilar mossy cells that drive inhibition in the DG and CA3 might also be a cause of breakdown of the DG gate.

Line 137. Presumably mossy fiber sprouting underlies the increase in connectivity from mature GCs to new GCs, but this is not explicitly addressed or discussed.

Are the results in Fig 3 consistent with recent results showing changes in synaptic activity in birth-dated GCs (Althaus et al., 2019)? – this should be discussed.

Another limitation of the approach is that the functional significance of altered dendrite angle is unclear.

Point-by-point Response

Reviewer #1:

1. Silencing of 2 wpi and 9 wpi GCs. The authors expressed DREADD receptors in a cohort of GCs born 1-2 days before pilocarpine injection. These adult born granule cells (abGCs) were silenced at different time points afterwards. Fig2. shows that silencing a cohort the abGC directly after pilocarpine injection for 2 weeks, reduces SRS frequency to about 40%, while the SRS duration is unaffected. By contrast silencing the abGCs 9 weeks after pilocarpine (Fig. 3) does not affect seizures at all. This is also consistent with the previous observation by the same authors, showing that complete ablation of adult neurogenesis does not affect seizure generation (Cho et al. 2015, Fig. 6). This indicates that abGCs may under some conditions potentially affect downstream hippocampal neurons generating the seizures, but are not themselves included in seizure generation. To understand the mechanisms underlying this phenomenon it would be important to study postsynaptic targets of the abGCs and how hyperactive abGCs can change downstream circuits. However, this was not investigated. As a first step, one could measure cFOS expression in downstream cells +/- CNO-mediated silencing or after Chr2-mediated stimulation of abGCs

We completely agree with the reviewer that abGC may affect downstream hippocampal neurons in epilepsy. To address this comment we have added experiments as they suggest to look at cFos expression in both hM3Dq and hM4Di conditions (Fig. S10) and elaborated in the discussion on the possibility of output changes and the importance of future studies to investigate this (see lines 308-316).

2. Activating abGCs. Conversely, Fig. 1 shows that CNO-mediated activation of a cohort of abGC for about 1 week can trigger the development of spontaneous seizures. However, this EEG activity is much shorter (13s, Fig. 1K) than after pilocarpine (30s, Fig 2LM). Also for 0-1w CNO there is no duration given in the table – why?

We thank the reviewer for pointing out this oversight. The seizure duration has been added to the table (Fig. 1K).

How do we know that the CNO-induced activity is the same as the pilocarpine-induced activity? Most importantly, the downstream synaptic modifications of CNO activation of abGC was not investigated. How do the mossy fibers look like? Does the size or the number of MFs increase? What about filopodial extensions in CA3?

We agree with the reviewer that downstream synaptic changes are likely occurring with CNO activation. From our previous work ablating abGCs, we concluded that ablation of aberrant neurogenesis does not alter mossy fiber sprouting. And we see no mossy fiber sprouting via ZNT3 staining in our activating abGC data (Fig. 1). To further investigate this we measured the change in ZNT3 staining in our mice from Fig. 2 that received pilocarpine and hM4Di + CNO and again saw no change in ZNT3 staining (Fig. S9) further supporting that aberrant abGCs do not contribute to changes in mossy fibers (see line 304-307). We were unable to measure filopodial extensions in the CA3 with our hM3Dq-gfp and hM4Di-gfp viruses as the GFP expression does not adequately fill the axon and agree with the reviewer they would be interesting experiments to conduct in future directions.

3. Dentate function. On a similar note, the authors claim that the dentate constitutes an inhibitory feedback circuit. However, the recurrent feedback inhibition does not exist in 1-2 week old cells, but develops well after 4 weeks post mitosis (Temprana et al. 2015). **By contrast feedforward excitation and**

inhibition in CA3 develops much earlier and could contribute to the effects observed by Hsieh and colleagues. Only considering feedback inhibition is too simplistic.

This is a great point. Because our strongest changes were in the morphology (dendrites) and migration of the abGCs, our initial focus was to examine changes in the inputs. We agree that the output changes is an important aspect and we're not excluding the feedforward excitation and inhibition in CA3 at earlier time points and this could contribute to the observed effects. We are developing tools to investigate the earliest output changes, such as feedforward connection to CA3 and feedback connection to mossy cells for future studies.

4. Upstream connectivity. Fig. 3 shows some synaptic inputs to abGCs after pilocarpine induced seizures or after CNO treatment. How does this compare with healthy controls? Why we don't see mossy cell inputs to abGCs? Fig. 4a need to be shown at much higher resolution. Nothing can be seen at present.

We have included sham data to compare to healthy controls. Here we do see mossy cells as identified by GluR2 staining. The lack of mossy cells in pilo, we argue, is due to the death after pilocarpine. While we see GluR2 positive cells in the pilo sections, none that co-localize with the input cells (**Fig. 4; lines 164-168**). Also we've included better representative images for Fig. 4.

5. Pilocarpine model. The pilocarpine model is special. How do the present results relate to other epilepsy models as for example the kainite model? The authors imply that pilocarpine is representative for all TLE.

We agree with the reviewer that the pilocarpine model is special. While there are other models (e.g. kainic acid, amygdala kindling, intrahippocampal kainic acid), we have found the pilocarpine model to be the most robust and reliable for generating spontaneous seizures while leaving the hippocampus, and specifically the dentate gyrus, intact. But we recognize that it does not represent all TLE. Therefore, we have changed any language that suggest pilocarpine represents all TLE.

Minor

1. How does the distribution of dendrites in Fig 2 JK compare to control? What about quantitative analysis fitting a gaussian distribution? Skewness of distribution in Fig. 2G could be quantified with fitting a skewed function to the distribution.

We thank the reviewer for pointing this out. The control distribution of dendrites is in Supplemental Fig. 2. We have also added statistics regarding test for normality using the D'Agostino & Pearson test to Figure 2 and Supplemental Figure 2.

2. What does the table in Fig. 3j represent? No comment in legend.

We thank the reviewer for pointing out this oversight. The appropriate text has been added to the figure legend.

3. Fig. 4 panel labeling does not correspond to legend.

The labeling for figure 4 has been correct in text and in the figure legend.

Reviewer #2 (Remarks to the Author):

We thank the reviewer for the interest they expressed and the constructive suggestions. Below is a point-by-point commentary:

The authors indicated that silencing early neuronal activity of newborn neurons restores proper morphology. This suggests that immature newborn neurons receive neuronal activity from surrounding neurons, affecting morphology when they become mature neurons. Presumably, this isn't only true after seizures but it seems rational to assume that newborn neurons produced in physiological conditions also receive neuronal activity from surrounding neurons. **Thus, it seems reasonable to examine the differences in the neuronal activity of newborn neurons produced in physiological conditions vs after seizures and to determine whether newborn neurons produced after seizures receive excessive neuronal activity / inputs than controls.** From their data and manuscript, I interpreted that newborn neurons receive excessive neuronal activity after seizure, thus they tried to repress it. Also Line 68 says "These overexcited interneurons thus impose a hyperexcitable signal to mature new neurons". However, unless I am missing the critical reference there isn't a clear message in the literature that immature neurons receive excessive neuronal activity from GABAergic neurons or other local neurons within 2 weeks from the born.

1) Perhaps the authors can show some evidence that newborn neurons after seizure have increased neuronal activity / more inputs from surrounding neurons in the critical period, by either citing the relevant papers or adding some additional experiments.

We appreciate that the reviewer astutely pointed out that the direct evidence is missing in linking activity of newborn neurons after pilocarpine to GABAergic neurons. To address this we included a number of additional experiments, two additional main figures, and have elaborated in the discussion on relevant papers that we think will address this concern. **See Fig. 5 and Fig. 6, lines 318-361.**

2) It would be useful to more clearly associate the abnormal morphology of immature neurons and the influences on their connectivity during early stages. The authors showed that newborn neurons produced after seizure have abnormal morphology when they become mature. However, the authors did not show whether this morphological abnormality is already detectable from early stages (during the 2 weeks after seizure). **This would help to establish whether the abnormal morphology is induced by abnormal activity or whether the activity induces the morphology.**

This is a very interesting question posed by the reviewer. To address this we added new experiments looking at migration and dendrite morphology in 2 week old adult-born GCs to determine if our morphology changes can be observed. At two weeks, we see that there are ectopic granule cells and a shift in the dendrite angle, however hM4Di silencing does not reduce these phenotypes, like it did in at a later time, suggesting they are only present beyond 2 weeks. We argue that the abnormal activity drives abnormal morphology. **Please see Fig. S4 and lines 116-122 and 272-276.**

3) Relate to the question 2, they showed that early neuronal activity of newborn neurons affects their morphology. However, they only showed the data at 8/9 weeks after CNO injection, and there is no data about the morphology of newborn neurons before they become mature. About this experiment, I am wondering if they can also address if morphological differences are observed immediately after 2 weeks treatment of CNO

Please see response above to number 2.

4) In figure 4, they injected two retroviruses into the DGs (one expresses TVA, the other is for hM4Di). However, both viruses express GFP. Thus, using only GFP expression, the author cannot distinguish if the cell is carrying both TVA and hM4Di. If they used only GFP to detect starter cells in the hM4Di group, their analysis would contain the cells that carry only TVA, but not hM4Di. I am wondering if they could show how many GFP+ cells (what percent of GFP+ cells) in hM4Di group have both TVA and hM4Di.

We thank the reviewer for raising this point. We have added additional images to Fig. 4 to clarify the expression between the cytoplasmic and nuclear GFP and to better illustrate the labeling of this system (see Fig. 4B and C). Further, we have elaborated more in the text that only double GFP cells were included in the analysis and that with the equal viral titers being simultaneously injected, we do not see many cells expressing only nuclear or cytoplasmic GFP only (see lines 148-152).

5) The authors showed that silencing early neuronal activity restores proper neuronal circuitry in seizure-induced newborn neurons. However, they did not show the neuronal circuit of newborn neurons in the physiological condition. This makes it difficult to interpret if the silencing early neuronal activity really contributes to restoring the proper neuronal circuit or yet a distinct type of circuit. If they say “restore”, it would be helpful to show the neuronal circuit of adult-born neurons in physiological conditions with/without silencing of neuronal activity.

We thank the reviewer for this oversight. To address this we have added these experiments to Fig. 4 and have expanded the text. See lines 142-176 and 293-304.

6) The authors proposed a critical period in which increased neuronal activity influences the morphology of newborn neurons, resulting in abnormal neuronal circuit. In figure 4, they identified the neuronal circuit of these seizure-induced neurons. However, given that silencing early neuronal activity can restore the proper neuronal circuit, **there must be some local neurons projecting to immature newborn neurons during the critical period, and these neurons are responsible for the increasing neuronal activity and changing the morphology of newborn neurons after seizure.** They assumed GABAergic interneurons give a tonic signal during the critical period. If so, I am wondering if they could perform similar rabies virus experiments during 2 weeks after pilocarpine treatment to identify the neurons projecting to the immature newborn neurons, and also if they could ask whether the subtype of these projecting neurons are different between the physiological conditions and after seizures.

The reviewer makes a very interesting point. The tonic GABA signal that is being detected in 2 week old adult-born GCs is not a synaptic GABA signal. We recognize this point may not have been emphasized, so we elaborated on the literature that demonstrates that functional synaptic connectivity does not appear until at least 3 weeks. Others have performed rabies tracing at early stages and only until after 15dpi does input appear (Vivar et al. 2012 Nat Comm doi: 10.1038/ncomms2101).

However, to better understand the increased activity in immature new neurons, we performed additional experiments using retroviral GCaMP6f and GABA pharmacology to demonstrate the increased role of GABA after seizures (See Fig. 5, and lines 177-213 and 318-361).

Reviewer #3

Line 54. The authors conclude that “excessive activity early in GC maturation generates aberrant neurons that have abnormal migration.....” and they use the terms activate and suppress to describe the DREADD manipulation. ***Excitatory and inhibitory DREADDs are typically used to increase and decrease spiking in neurons, so it is unclear they affect cells that may not fire action potentials (i.e. progenitors).*** In order to understand the mechanism by which adult-born GCs undergo abnormal maturation and the extent to which chemogenetics mimic extrinsic cues during epileptogenesis, ***the authors should determine how their CNO “activates” or “suppresses” retroviral-labeled cells.***

The reviewer brings up a very important point that immature neurons adult-born granule (<3 weeks of age) do not spontaneously fire action potentials. Thus it is currently unknown how DREADDs affect these cells at this age. To address this, we include a number of additional experiments that link immature adult-born GC function to intracellular calcium. We have added new data that demonstrates calcium activity is altered by DREADD activity, both to increase and decrease it accordingly. **See Fig. 5 and 6 and lines 177-236 and 318-362.**

A related point is that there is no justification for the duration of the CNO treatment regime – was one week necessary to see an effect? If chemogenetic activation of new cells mimics extrinsic cues that drive abnormal migration, connectivity and SRS after SE, *then the pathological changes would likely be triggered by a shorter treatment of CNO that lasts the duration of status epilepticus.*

We recognize our rationale was unclear for CNO treatment and have clarified in the text. We agree to mimic SE a shorter duration of CNO would be appropriate, however we were modeling the persistent GABA signal that SE generates. Because this is a long-term change in the hippocampal niche, we chose a long-term treatment regime. **See lines 71-77.**

Minor issues:

Figure 1K. The seizure duration seems to be absent from the 0-1w group in the Table.

We have corrected the oversight and added the seizure duration.

At several places in the manuscript the authors focus on the DG as an inhibitory feedback circuit and the putative role of adult-born neurons in feedback inhibition. However, this focus on feedback inhibition is not supported by citations, and is a little distracting since there are several forms of inhibition that maintain gating function. More importantly, ***very recent results seem to rule out a potential role of newborn neurons in conventional feedback inhibition mediated by GABAA receptors (in favor of an unconventional form of direct inhibition, Luna et al., 2019).*** Since the current results don’t address the role of newborn cells in inhibitory function, the authors might want to tone down this focus.

We thank the reviewer for bringing this to our attention. We do not intend to claim that abGCs establish the inhibitory feedback circuit itself, rather after pilocarpine, changes with the abGC population disrupt the inhibitory circuit. As the reviewer highlights, inputs from the entorhinal cortex (which we see as a major change in our pilo model) provides a bidirectional modulation of the DG circuits. We have included this in our discussion **(lines 293-298)** and changed or toned down any language that suggests the newborn cells play an inhibitory role.

Line 66. Presumably it is meant that PV and SST are over-excited before they die. Regardless, there is conflicting data about what happens to particular interneuron subtypes in mouse epilepsy models (i.e. Huusko et al., 2015; Buckmaster et al., 2017).

We have expanded on this in the text and included alternative possibilities for interneuron subtypes in epilepsy models. **Lines 71-74.**

Line 82. Since no dendrite abnormalities were detected, this text is confusing, “To determine if abnormal migration and dendrite changes...”

We have clarified the text

Line 125. Loss of hilar mossy cells that drive inhibition in the DG and CA3 might also be a cause of breakdown of the DG gate.

We have included further discussion about mossy cells. **See lines 164-168.**

Line 137. Presumably mossy fiber sprouting underlies the increase in connectivity from mature GCs to new GCs, but this is not explicitly addressed or discussed.

We have added data looking at MFS. **See Supplemental figure 9.**

Are the results in Fig 3 consistent with recent results showing changes in synaptic activity in birth-dated GCs (Althaus et al., 2019)? – this should be discussed.

We do believe our results are consistent with the heterogeneous nature of aberrant granule cells and that EGCs form an epileptic hub network, similarly discussed in Althaus 2019. We extensively discussed this concept throughout the discussion and specifically in lines **262-264.**

Another limitation of the approach is that the functional significance of altered dendrite angle is unclear.

We have elaborated on the significance of dendrite angle in the text. **See lines 107 and 290-291**

Reviewers' Comments:

Reviewer #1:

Remarks to the Author:

Although I can see some improvement, many of my concerns are still valid. As a major problem, there is still no cellular mechanism. If this does not improve, the study is only correlative and the claims should be toned down substantially. There are some new calcium imaging experiments in 2-week-old granule cells. Due to conceptual and analysis problems with the life-cell imaging, these data need to be reanalyzed (see below).

1. Are young GCs necessary for seizures? On page 3, 3rd para, the authors write that "early intrinsic activity is sufficient and necessary for cell intrinsic pathological changes and recurrent SRS development". This is not true, because as shown in Fig. 2M seizure activity is only reduced to about 50%. That means there are seizures without activity in the young cells, i.e. activity in these cells is not necessary for seizure development. Instead, the data are consistent with a modulatory role of young GCs in seizure development in cooperation with some unknown seizure-generating network (maybe CA3).

2. Morphological changes. Early activity during the first 2 weeks does not induce morphological changes in GCs (Fig. S1, Fig. S4). Just at 8-9 wpi changes are visible. How do we know that the morphological changes of GCs at 9 wpi are caused/induced by activity in 1-2 wpi cells? I think the data would be fully consistent with pilocarpine (or CNO) inducing pro-epileptic changes in the Hippocampus and subsequent seizure activity in multiple hippocampal subnetworks affect dendritic morphology in maturing GCs (completely unrelated to early activity in 1wpi cells). That would also mean that the morphology could be changed in older or younger granule cells, not directly involved in the early activity at 1-2wpi. Can we exclude this possibility? If not the authors should clearly state that there is no evidence that morphological changes in GCs are causally related to seizures. Instead the morphology would be an epiphenomenon.

3. Calcium imaging. The authors provide completely new data using the calcium sensor GCamp6f in acute hippocampal slices, which is nice in general. However, they use a strange way to analyze the data. According to methods, they measure the fluorescence in a ROI covering a single granule cell (F) as well as Background fluorescence next to the cell (BG). They pretend to calculate dF/F . However, instead they measure $(F(t)-BG)/BG$, basically corresponding to the Background subtracted Raw GCamp6 fluorescence. Furthermore, they classify any fluorescence change larger than BG as a calcium event / transient (methods page 3, 3rd para). This is strange and completely useless, because any noisy baseline fluorescence would be falsely classified as an event, which appears to be the case Fig. 5O, Fig 6C, Fig. 6E and Fig. 6M (CNO). For example, Fig. 5O is probably just noisy baseline fluorescence and the number of events in Bicuculine (Fig. 5L) is probably equal to zero.

The authors need to get an estimate of the real baseline fluorescence F_{base} in the cells, which is larger than background. Afterwards, dF/F is calculated as $(F(t) - F_{base}) / (F_{base} - BG)$.

I understand that this can be difficult with spontaneously active cells. Sometimes it helps to generate all-data-histograms. If there is a peak on the left, this peak typically corresponds to baseline fluorescence. Fitting the distribution will also reveal the SD of noise fluctuations, which will allow to define an event as fluorescence exceeding baseline more than 2 times SD.

Alternatively, baseline fluorescence could be measured with synaptic blockers.

4. Tonic GABA. The authors measure spontaneous calcium transients in 2wpi GCs (Fig. 5F, H, N, P) which is nice. They conclude this is generated by tonic GABA. They completely ignore, that in 2 wpi cells there is a large complement of functional GABAergic synapses formed by neurogliaform cells (Markwardt et al. 2011, Nat Neurosci 14:1407), PV basket cells and Somatostatin positive HIPP cells (Alvarez et al. 2016, Science 354:459, Groisman et al. 2020, Cell Rep 30:202). Similarly, at 1-2 wpi, there are glutamatergic synapses with dominating NMDA receptor contribution (Sah et al. 2017 Scientific Reports 7:10903, Li et al. 2017 Elife 6:e23612). Also it was shown that

depolarizing GABAergic synapses can efficiently evoke AP firing at 2 wpi granule cells (Heigele et al. 2016, Nat Neurosci 19:263) which can generate nice and large AP-evoked calcium transients in granule cells younger than 3 wpi (Stocca et al. 2008 J Physiol 586:3795).

Therefore, the most straightforward interpretation of the data is that the calcium transients (Fig. 5F, H, N, P and Fig. 6M washout) are evoked by phasic synaptic transmission.

The authors should try to separate phasic from tonic inhibition with application of 1 μ M TTX, which should leave tonic GABA untouched. Furthermore, the contribution of NMDA-synapses could be tested with 50 μ M AP5.

5. Discussion'. In the discussion (page 8), the authors write "Functional synapses, both GABA and glutamatergic, do not form until around 3 weeks and only small amplitude suprathreshold action potentials can be detected". As explained above this is wrong and should be corrected by citing the relevant literature.

6. Conclusions in abstract: In case there will be no mechanisms, the wording in the abstract has to be changed:

- we identified a critical window of activity that "is associated with" (instead of drives) aberrant maturation... (because DREADD activation of adult-born GCs does not alter dendritic morphology (Fig. S1) and because silencing of 1-2 wpi cells in pilo does not change morphology of these cells (Fig. S4) although it reduced SRS-development (Fig. 2M)).

- Silencing of a restricted cohort..."reduced" (instead of prevented) abnormal development and SRS (because there is still about 50% of seizure activity left (Fig. 2M)).

- please delete "thereby restoring the protective function back to the dentate gyrus" (because there is no supporting data for this claim)

Minor

7. Page 4 3rd para. "Silencing aberrant neurogenesis prevents.." should be replaced with "silencing young neurons after pilocarpin treatment prevents..." because not neurogenesis was silenced, but the activity in young neurons.

Reviewer #2:

None

Reviewer #3:

Remarks to the Author:

The authors have added significant new data to address the majority of the reviewer concerns. I find the original conclusions to be strengthened by the new data (Figs 1-4, 6). However, the new imaging data in Fig 5 is too preliminary to support the conclusions about GABA-mediated Ca²⁺ influx in abGC abnormal migration. Rigorously addressing the mechanisms leading to abnormal migration seems beyond the scope of the current project.

Major Points

Figure 5. While there appear to be differences in spontaneous Ca²⁺ influx between abGCs in sham and pilo conditions, the authors have not determined the source of the Ca²⁺ transients that could include synaptic depolarization, release from intracellular stores or action potentials, among others (reviewed by Grienberger and Konnerth, 2012). Pharmacology could be used to sort these out,

and to isolate Ca²⁺ responses resulting from GABA_A depolarization. But the authors did not perform the experiments that would be needed to compare GABA_A-mediated Ca²⁺ transients in sham and pilo (GABA was tested in sham, bic was tested in pilo), so they cannot make any conclusions about differences in GABAergic signaling in sham and pilo. The raw data in Fig 5N,O,P and the video does suggest that the largest Ca²⁺ events are mediated by GABA_A receptors, but the lack of effect on event frequency (Fig 5K) suggests there are other sources, as well. It is also unclear why the authors interpret the results as tonic/ambient GABAergic signaling, since many studies have shown that by two weeks of age abGCs respond to synaptically-released GABA from at least two classes of interneurons (Esposito et al., 2005; Overstreet Wadiche et al., 2005; Markwardt et al., 2009; 2011; Song et al., 2013; Alvarez et al., 2016; Groisman et al., Cell Reports 2020; Vaden et al. eLife, 2020). Regardless, the lack of pharmacological characterization of the Ca²⁺ signals make the imaging data too preliminary to support the conclusions about GABAergic signaling (lines 335-360 and elsewhere). I think Figure 5 and related text should be removed or additional experiments/analysis performed to strengthen it. The authors might speculate on the potential role of GABAergic depolarization/Ca²⁺ signaling contributing to abnormal migration in the discussion.

The data in Figure 6 suggests that DREADD activation modulates Ca²⁺ signaling. The authors should provide more information about how Ca²⁺ signals were analyzed and acknowledge that there are multiple sources of Ca²⁺ signaling that could be modified by CNO. If the authors want conclude that CNO specifically modifies GABA_A-mediated Ca²⁺ signaling, they would need to show that CNO has no effect on Ca²⁺ signals in the presence of a GABA_A antagonist, or show that CNO modulates pharmacologically-isolated GABA_A-mediated Ca²⁺ signals.

Minor Points

1. Related to Rev 1 point 3 and Rev 3 minor point 2. Since the authors acknowledge feedforward excitation and inhibition of CA3 is a major aspect of dentate function, they might want to further tone down the first few sentences of the abstract that focuses exclusively on feedback inhibition.
2. Line 137. Since their data does not show major changes in PV and SST circuitry, the authors might want to delete this statement (or support it with references).
3. Figure 6L, it looks like CNO reduced the baseline Ca²⁺ level rather than suppressing Ca²⁺ transients. It would be useful to label Ca²⁺ transients in the raw traces. Please provide more details about sampling rate and exposure times, and number of cells labeled versus number of cells exhibiting transients. It seems like the low sampling rate used in the Ca²⁺ imaging experiments could preclude accurate detection of fast Ca²⁺ transients.

REVIEWER COMMENTS

Reviewer #1 (Remarks to the Author):

Although I can see some improvement, many of my concerns are still valid. As a major problem, there is still no cellular mechanism. If this does not improve, the study is only correlative and the claims should be toned down substantially. There are some new calcium imaging experiments in 2-week-old granule cells. Due to conceptual and analysis problems with the live-cell imaging, these data need to be reanalyzed (see below).

1. Are young GCs necessary for seizures? On page 3, 3rd para, the authors write that “early intrinsic activity is sufficient and necessary for cell intrinsic pathological changes and recurrent SRS development”. This is not true, because as shown in Fig. 2M seizure activity is only reduced to about 50%. That means there are seizures without activity in the young cells, i.e. activity in these cells is not necessary for seizure development. Instead, the data are consistent with a modulatory role of young GCs in seizure development in cooperation with some unknown seizure-generating network (maybe CA3).

We agree with the reviewer it is important to demonstrate causality regarding the role of adult-born GCs (abGCs) and seizures. The reviewer is correct to point out that while hM4Di silencing of 0-2 week old abGCs is sufficient to reduce seizures by ~50%, there are still seizures in pilocarpine-treated animals. This reduction in SRS frequency is comparable to previous work from our lab and others using genetic ablation models¹⁻⁴. We do not yet know if abGCs only contribute or modulate seizures as the reviewer suggests or if this is a caveat of the pilocarpine status epilepticus model. We believe our data using the hM3Dq activation of 0-2 week old abGCs in a non-epileptic model strongly demonstrates that this early activity is sufficient to promote seizures. We agree with the reviewer about necessity and have toned down the language to reflect the modulatory role. **(see Page 3 lines 113-115)**. We also believe that future work addressing the role of abGCs in seizure generation particularly in non-status models would be important in the field and beyond the scope of this work.

2. Morphological changes. Early activity during the first 2 weeks does not induce morphological changes in GCs (Fig. S1, Fig. S4). Just at 8-9 wpi changes are visible. How do we know that the morphological changes of GCs at 9 wpi are caused/induced by activity in 1-2 wpi cells? I think the data would be fully consistent with pilocarpine (or CNO) inducing pro-epileptic changes in the Hippocampus and subsequent seizure activity in multiple hippocampal subnetworks affect dendritic morphology in maturing GCs (completely unrelated to early activity in 1wpi cells). That would also mean that the morphology could be changed in older or younger granule cells, not directly involved in the early activity at 1-2wpi. Can we exclude this possibility? If not the authors should clearly state that there is no evidence that morphological changes in GCs are causally related to seizures. Instead the morphology would be an epiphenomenon.

We thank the reviewer for pointing out a key negative result that we did not focus on. In this paper, we define dendrite morphology to include both dendritic complexity and angle of the primary dendrite angle. At the early time point (0-2wks), the only change we see in dendrite complexity is shown in **Fig. S3C** when hM4Di + CNO was used to silence cells in a sham condition. At later time points, we did not observe changes to dendrite complexity in any other condition. For dendritic angle, the most drastic change was observed between sham and pilo (**Fig. S2D**) and when we used hM4Di +CNO in pilo model (**Fig. 2K**). We also did not observe changes in dendritic angle at later time points in any condition.

We agree with the reviewer that activity during the first 2 weeks does not induce changes in dendrite complexity, but it does induce changes in dendrite angle. In Supplemental Figure 4, we look at morphology immediately after 2 weeks and only observe changes between 2 week old GCs from sham and pilo in primary angle. This suggests that even as early as 2 weeks, the primary dendrite is angled abnormally. However, we do not observe a normalization of the angle at 2w with hM4Di+CNO, which suggests that those changes we see once in mature GCs occur later. We agree with the reviewer that we cannot exclude these change in dendrite angle as an epiphenomenon. That is, we cannot exclude that other neighboring GCs are causing the change in dendrite angle. Thus, we have removed any wording suggesting a causal link between morphology and seizures. **See lines 85-87, 93-95, and 114-115.**

3. Calcium imaging. The authors provide completely new data using the calcium sensor GCamp6f in acute hippocampal slices, which is nice in general. However, they use a strange way to analyze the data. According to methods, they measure the fluorescence in a ROI covering a single granule cell (F) as well as Background fluorescence next to the cell (BG). They pretend to calculate dF/F . However, instead they measure $(F(t) - BG)/BG$, basically corresponding to the Background subtracted Raw GCamp6 fluorescence. Furthermore, they classify any fluorescence change larger than BG as a calcium event / transient (methods page 3, 3rd para). This is strange and completely useless, because any noisy baseline fluorescence would be falsely classified as an event, which appears to be the case Fig. 5O, Fig 6C, Fig. 6E and Fig. 6M (CNO). For example, Fig. 5O is probably just noisy baseline fluorescence and the number of events in Bicuculline (Fig. 5L) is probably equal to zero. The authors need to get an estimate of the real baseline fluorescence F_{base} in the cells, which is larger than background. Afterwards, dF/F is calculated as $(F(t) - F_{base}) / (F_{base} - BG)$. I understand that this can be difficult with spontaneously active cells. Sometimes it helps to generate all-data-histograms. If there is a peak on the left, this peak typically corresponds to baseline fluorescence. Fitting the distribution will also reveal the SD of noise fluctuations, which will allow to define an event as fluorescence exceeding baseline more than 2 times SD. Alternatively, baseline fluorescence could be measured with synaptic blockers.

We thank the reviewer for their positive comments regarding our new GCamp6f imaging data and thank the reviewer for suggesting a method to calculate calcium activity. Using this method, we have reanalyzed all of the previous calcium imaging data along with the new data include in the submission to include the baseline subtraction. We have also improved our definition of event detection to include rise time and 2 standard deviations above baseline. Please see the revised **Fig. 5, Fig. 6, and Fig. S8, and Fig. S9 (lines 188-237) and methods 109-113.**

4. Tonic GABA. The authors measure spontaneous calcium transients in 2wpi GCs (Fig. 5F, H, N, P) which is nice. They conclude this is generated by tonic GABA. They completely ignore, that in 2 wpi cells there is a large complement of functional GABAergic synapses formed by neurogliaform cells (Markwardt et al. 2011, Nat Neurosci 14:1407), PV basket cells and Somatostatin positive HIPP cells (Alvarez et al. 2016, Science 354:459, Groisman et al. 2020, Cell Rep 30:202). Similarly, at 1-2 wpi, there are glutamatergic synapses with dominating NMDA receptor contribution (Sah et al. 2017 Scientific Reports 7:10903, Li et al. 2017 Elife 6:e23612). Also it was shown that depolarizing GABAergic synapses can efficiently evoke AP firing at 2 wpi granule cells (Heigele et al. 2016, Nat Neurosci 19:263) which can generate nice and large AP-evoked calcium transients in granule cells younger than 3 wpi (Stocca et al. 2008 J Physiol 586:3795).

Therefore, the most straightforward interpretation of the data is that the calcium transients (Fig. 5F, H, N, P and Fig. 6M washout) are evoked by phasic synaptic transmission.

We thank the reviewer for pointing out that calcium transients can be generated by both tonic and synaptic sources of GABA in cells 2 wpi. We have performed a new set of experiments comparing the calcium activity of sham vs pilo in 2 week old cells (**Fig. 5**). In this set of experiments we included TTX to block AP-evoked transients and bicuculline to block GABA_A. Following TTX, we see the presence of longer, sustained events. In the text we described them as reminiscent of tonic GABA because once bicuculline was applied it abolished all responses; both the long, sustained events, but also the short, faster events. In sham mice, we observe that TTX decreases the amplitude of these events suggesting a contribution of APs possibly similar to those previously measured by others^{5,6}. When we add bicuculline, pretty much all events are eliminated. This, however, is not true in the pilo group. In pilo, we observe an elevated baseline but application of TTX appears to have little effect, suggesting that the enhanced calcium baseline is not associated with increased APs. Application of bicuculline appears to significantly lower baseline calcium levels, but interestingly does not reduce the events to the extent we see in sham.

We thank the reviewer for pointing out the early contribution of glutamate in cells 2wpi. This was a possible explanation to the remnant calcium events observed in the pilo group following TTX and bicuculline. However, application of AP5/CNQX to block the ionotropic glutamate receptors, NMDA and AMPA, had a very modest

effect on calcium activity (Fig. S9; see lines 225-237). We have changed the text to reflect this modified interpretation (see lines 188-207).

The authors should try to separate phasic from tonic inhibition with application of 1 μ M TTX, which should leave tonic GABA untouched. Furthermore, the contribution of NMDA-synapses could be tested with 50 μ M AP5.

Thank you for this suggestion. Our new data in the new Fig. 5 and Fig. S8 has some interesting results pertaining to these experiments.

5. Discussion. In the discussion (page 8), the authors write "Functional synapses, both GABA and glutamatergic, do not form until around 3 weeks and only small amplitude suprathreshold action potentials can be detected". As explained above this is wrong and should be corrected by citing the relevant literature.

We have removed that sentence.

6. Conclusions in abstract: In case there will be no mechanisms, the wording in the abstract has to be changed:

-We identified a critical window of activity that "is associated with" (instead of drives) aberrant maturation... (because DREADD activation of adult-born GCs does not alter dendritic morphology (Fig. S1) and because silencing of 1-2 wpi cells in pilo does not change morphology of these cells (Fig. S4) although it reduced SRS-development (Fig. 2M)).

-Silencing of a restricted cohort..."reduced" (instead of prevented) abnormal development and SRS (because there is still about 50% of seizure activity left (Fig. 2M)).

-Please delete "thereby restoring the protective function back to the dentate gyrus" (because there is no supporting data for this claim)

We thank the reviewer for these suggestions and have made these changes.

Minor

7. Page 4 3rd para. "Silencing aberrant neurogenesis prevents.." should be replaced with "silencing young neurons after pilocarpin treatment prevents...." because not neurogenesis was silenced, but the activity in young neurons.

We have made this change.

Reviewer #2 (Remarks to the Author):

The authors have done an admirable job of answering the issues that I raised.

We thank the reviewer for the positive comments.

Reviewer #3 (Remarks to the Author):

The authors have added significant new data to address the majority of the reviewer concerns. I find the original conclusions to be strengthened by the new data (Figs 1-4, 6). However, the new imaging data in Fig 5 is too preliminary to support the conclusions about GABA-mediated Ca²⁺ influx in abGC abnormal migration. Rigorously addressing the mechanisms leading to abnormal migration seems beyond the scope of the current project.

Major Points

Figure 5. While there appear to be differences in spontaneous Ca²⁺ influx between abGCs in sham and pilo conditions, the authors have not determined the source of the Ca²⁺ transients that could include synaptic depolarization, release from intracellular stores or action potentials, among others (reviewed by Grienberger

and Konnerth, 2012). Pharmacology could be used to sort these out, and to isolate Ca²⁺ responses resulting from GABA_A depolarization. But the authors did not perform the experiments that would be needed to compare GABA_A-mediated Ca²⁺ transients in sham and pilo (GABA was tested in sham, bic was tested in pilo), so they cannot make any conclusions about differences in GABAergic signaling in sham and pilo. The raw data in Fig 5N,O,P and the video does suggest that the largest Ca²⁺ events are mediated by GABA_A receptors, but the lack of effect on event frequency (Fig 5K) suggests there are other sources, as well.

We agree with the reviewer that it is important to determine the source of the calcium transients. We have now performed additional new experiments comparing GABA mediated calcium transients in sham and pilo, (**see Fig. 5; lines 188-209**). We have included TTX to block AP-evoked calcium activity and bicuculline to block GABA_A receptors, those previously reported in regulating 2 week old cell. At baseline, we observe short duration, fast transients. TTX appears to unmask longer, sustained events. Both types of events are abolished with bicuculline suggesting they are both dependent on GABA_A activation.

Interestingly, this was different in pilo. Calcium levels were elevated at baseline to begin with and TTX did little to alter that response. Bicuculline did abolish most of the calcium response, but a few smaller calcium events remain. This small calcium remnant was not due to ionotropic glutamate, as we also applied AP5/CNQX to block NMDA and AMPA receptors and observed little effect (**Fig. S9; lines 225-237**). We cannot exclude the possibility of a metabotropic contribution. Future studies are required to identify this source of this small calcium contribution.

It is also unclear why the authors interpret the results as tonic/ambient GABAergic signaling, since many studies have shown that by two weeks of age abGCs respond to synaptically-released GABA from at least two classes of interneurons (Esposito et al., 2005; Overstreet Wadiche et al., 2005; Markwardt et al., 2009; 2011; Song et al., 2013; Alvarez et al., 2016; Groisman et al., Cell Reports 2020; Vaden et al. eLife, 2020). Regardless, the lack of pharmacological characterization of the Ca²⁺ signals make the imaging data too preliminary to support the conclusions about GABAergic signaling (lines 335-360 and elsewhere). I think Figure 5 and related text should be removed or additional experiments/analysis performed to strengthen it. The authors might speculate on the potential role of GABAergic depolarization/Ca²⁺ signaling contributing to abnormal migration in the discussion.

We thank the reviewer for these suggestions. We have included the pharmacology as recommended and replaced Fig. 5 with all new data to address the source of calcium changes (**see lines 188-224**). With the additional experiments we can conclude the source of calcium transients are largely GABA_A receptor mediated.

The data in Figure 6 suggests that DREAAD activation modulates Ca²⁺ signaling. The authors should provide more information about how Ca²⁺ signals were analyzed and acknowledge that there are multiple sources of Ca²⁺ signaling that could be modified by CNO. If the authors want conclude that CNO specifically modifies GABA_A-mediated Ca²⁺ signaling, they would need to show that CNO has no effect on Ca²⁺ signals in the presence of a GABA_A antagonist, or show that CNO modulates pharmacologically-isolated GABA_A-mediated Ca²⁺ signals.

We agree with the reviewer it would be interesting, in future studies, to identify the sources of calcium that could be modified by the DREADDs. We speculate on key intracellular pathways that hM4Di and hM3Dq can possibly target to modulate calcium (**lines 335-339**). This does not exclude a direct or indirect activation of GABA_A receptor. Likely, the DREADDs and GABA are altering intracellular calcium via independent pathways. The pattern of activation with GABA (**Fig. S9**) and hM3Dq (**Fig. 6A**) have different kinetics and levels of activation. Application of exogenous GABA produces the long, sustained responses, whereas hM3Dq appears to increase the amplitude of the events, but they remain short in duration. This suggests separate mechanisms for calcium activation. These additional experiments to identify the link between DREADD activation and GABA would be interesting but beyond the scope of this work.

Minor Points

1. Since the authors acknowledge feedforward excitation and inhibition of CA3 is a major aspect of dentate

function, they might want to further tone down the first few sentences of the abstract that focuses exclusively on feedback inhibition.

We have modified this in the abstract.

2. Line 137. Since their data does not show major changes in PV and SST circuitry, the authors might want to delete this statement (or support it with references).

We thank the reviewer for pointing this out. While our data demonstrates no changes from PV and SST to adult-born granule cells, these cell populations have been demonstrated to play a protective function in the overall dentate circuitry. We have added the missing reference.

3. Figure 6L, it looks like CNO reduced the baseline Ca²⁺ level rather than suppressing Ca²⁺ transients. It would be useful to label Ca²⁺ transients in the raw traces. Please provide more details about sampling rate and exposure times, and number of cells labeled versus number of cells exhibiting transients. It seems like the low sampling rate used in the Ca²⁺ imaging experiments could preclude accurate detection of fast Ca²⁺ transients.

We thank the reviewer for the comments regarding the calcium imaging. We've included more details about image acquisition to include sampling rate and cells inclusion criteria in the methods (see **Methods lines 100-106**). We have also included traces from all cells (**Fig. 5, Fig. 6, Fig. S8, Fig.S9**).

References:

- 1 Cho, K. O. *et al.* Aberrant hippocampal neurogenesis contributes to epilepsy and associated cognitive decline. *Nat Commun* **6**, 6606, doi:10.1038/ncomms7606 (2015).
- 2 Varma, P., Brulet, R., Zhang, L. & Hsieh, J. Targeting Seizure-Induced Neurogenesis in a Clinically Relevant Time Period Leads to Transient But Not Persistent Seizure Reduction. *J Neurosci* **39**, 7019-7028, doi:10.1523/JNEUROSCI.0920-19.2019 (2019).
- 3 Pun, R. Y. *et al.* Excessive activation of mTOR in postnatally generated granule cells is sufficient to cause epilepsy. *Neuron* **75**, 1022-1034, doi:10.1016/j.neuron.2012.08.002 (2012).
- 4 Hosford, B. E., Rowley, S., Liska, J. P. & Danzer, S. C. Ablation of peri-insult generated granule cells after epilepsy onset halts disease progression. *Sci Rep* **7**, 18015, doi:10.1038/s41598-017-18237-6 (2017).
- 5 Esposito, M. S. *et al.* Neuronal differentiation in the adult hippocampus recapitulates embryonic development. *J Neurosci* **25**, 10074-10086, doi:10.1523/JNEUROSCI.3114-05.2005 (2005).
- 6 Heigele, S., Sultan, S., Toni, N. & Bischofberger, J. Bidirectional GABAergic control of action potential firing in newborn hippocampal granule cells. *Nat Neurosci* **19**, 263-270, doi:10.1038/nn.4218 (2016).

Reviewers' Comments:

Reviewer #1:

Remarks to the Author:

The paper has substantially improved. The only thing missing, is that the title should be adopted to the new interpretation of the data.

Instead of '...rewires hippocampus circuitry to CAUSE epilepsy' it should be '...rewires hippocampal circuitry to aggravate epilepsy'

Reviewer #3:

Remarks to the Author:

The authors have largely addressed my previous concerns. I just have a couple of minor suggestions they may want to consider.

Lines 247-250 needs some grammatical edits.

Line 330. I did not find Fig S12 to be a convincing approach to address the contribution of new GCs to seizure activity, whereas recent in vivo recordings during seizures (Sparks et al., Nat Commun 2020) support a slightly different view about the contribution of adult-born GCs that the authors might want to discuss.

Line 345. It might be informative to discuss recent evidence that likewise supports G-protein regulation of Ca²⁺ influx in new GCs as a mechanism that regulates maturation in a physiological rather than pathological context (Gao et al., Cell Reports, 2020).

REVIEWERS' COMMENTS

Reviewer #1 (Remarks to the Author):

The paper has substantially improved. The only thing missing, is that the title should be adopted to the new interpretation of the data. Instead of '...rewires hippocampus circuitry to CAUSE epilepsy' it should be '...rewires hippocampal circuitry to aggravate epilepsy'

We thank the reviewer for the positive comments. We have changed the title to reflect the new interpretation of the data and to match the editorial suggestions. The reviewer's suggestion of 'aggravate epilepsy' was included into the final changes of the abstract.

Reviewer #3 (Remarks to the Author):

The authors have largely addressed my previous concerns. I just have a couple of minor suggestions they may want to consider.

We thank the reviewer for all the helpful comments and suggestions.

Lines 247-250 needs some grammatical edits.

Thank you for highlighting this error. We have corrected the grammatical errors.

Line 330. I did not find Fig S12 to be a convincing approach to address the contribution of new GCs to seizure activity, whereas recent in vivo recordings during seizures (Sparks et al., Nat Commun 2020) support a slightly different view about the contribution of adult-born GCs that the authors might want to discuss.

We thank the reviewer for suggesting this citation. We think it supports our conclusion that aberrant GCs generate seizure hub networks in the dentate circuitry and we have included a brief discussion to highlight an alternative view on the contribution of abGCs.

Line 345. It might be informative to discuss recent evidence that likewise supports G-protein regulation of Ca²⁺ influx in new GCs as a mechanism that regulates maturation in a physiological rather than pathological context (Gao et al., Cell Reports, 2020).

Thank you for highlighting this citation from the Zhao and Overstreet-Wadiche labs. The physiological role of calcium and GCPRs on abGC maturation was included in a brief discussion point.